# Unravelling cytosolic delivery of cell penetrating peptides with a quantitative endosomal escape assay

Serena L. Y. Teo[1,2,3], Joshua J. Rennick [1,2,3], Daniel Yuen [1,2], Hareth Al-Wassiti[1], Angus P. R. Johnston [1,2✉] & Colin W. Pouton [1✉]

Cytosolic transport is an essential requirement but a major obstacle to efficient delivery of therapeutic peptides, proteins and nucleic acids. Current understanding of cytosolic delivery mechanisms remains limited due to a significant number of conflicting reports, which are compounded by low sensitivity and indirect assays. To resolve this, we develop a highly sensitive Split Luciferase Endosomal Escape Quantification (SLEEQ) assay to probe mechanisms of cytosolic delivery. We apply SLEEQ to evaluate the cytosolic delivery of a range of widely studied cell-penetrating peptides (CPPs) fused to a model protein. We demonstrate that positively charged CPPs enhance cytosolic delivery as a result of increased non-specific cell membrane association, rather than increased endosomal escape efficiency. These findings transform our current understanding of how CPPs increase cytosolic delivery. SLEEQ is a powerful tool that addresses fundamental questions in intracellular drug delivery and will significantly improve the way materials are engineered to increase therapeutic delivery to the cytosol.

[1] Monash Institute of Pharmaceutical Sciences, Monash University, Parkville, VIC, Australia. [2] ARC Centre of Excellence in Convergent Bio-Nano Science and Technology, Monash University, Parkville, VIC, Australia. [3] These authors contributed equally: Serena L. Y. Teo, Joshua J. Rennick. ✉email: angus.johnston@monash.edu; colin.pouton@monash.edu

Biological therapies such as peptides, proteins and nucleic acids have emerged as promising approaches for combating a wide variety of infectious, immunological and genetic disorders[1,2]. In order to elicit a therapeutic response, these macromolecules need to interact with their corresponding targets that are often located within the cell. However, unlike many low molecular weight drugs, they do not readily diffuse across cell membranes due to their large size. Instead, they are typically taken up into cells by endocytosis[3–5]. Through this pathway, most macromolecules remain trapped within membrane-bound endocytic vesicles, separating them from their required site of action. Thus, overcoming endosomal entrapment is vital for successful therapeutic delivery. However, the process of endosomal escape is inefficient and a major rate-limiting step in intracellular delivery of biological therapies[3,4].

To improve the delivery of biological therapies, cell-penetrating peptides (CPPs) have emerged as promising delivery agents for enhancing cytosolic delivery. Initial reports suggested that CPPs enter cells via direct translocation[6] across the plasma membrane, but subsequent re-evaluation studies indicate that uptake of CPPs occurs by endocytosis[7]. Following endocytic uptake, some CPPs are thought to gain entry to the cytosol by promoting endosomal membrane fusion and destabilisation[8,9] (referred to as endosomal escape peptides or EEPs). While this has been the widely accepted explanation, a number of reports have argued against the ability of EEPs to induce endosomal release after cellular uptake[10,11]. To date, there is no clear consensus on how EEPs mediate intracellular delivery[12,13]. This remains a critical issue that must be addressed in order to design more effective delivery vectors that promote endosomal release.

The uncertainty of endosomal escape mechanisms stems from the lack of methods that can quantify endosomal escape directly and reliably. Assays that measure biological activity of biomolecules are indirect approaches as they require a cascade of events to occur following endosomal escape (e.g., transcription/translation or nuclear translocation)[5,14]. In these assays, it is challenging to decouple endosomal escape from inefficiencies in these downstream processes.

Alternative methods rely on observing intracellular distribution of fluorescently labelled materials by fluorescence microscopy. While this approach provides a direct visualisation of subcellular localisation of materials, the main disadvantage is that it is challenging to observe weak, diffuse signal—indicating cytosolic delivery—in the presence of bright, punctate signal in endo/lysosomes. Furthermore, any punctate signal that does not colocalise with common endosomal markers such as Rab5 or EEA1 (early endosome), Rab7 (late endosome) and LAMP1 (lysosome) is often mischaracterised as endosomal escape. Co-incubation of materials of interest with calcein, a small membrane-impermeable dye that appears punctate when sequestered within endosomal/lysosomal compartments, is often another method used to overcome the requirement for fluorescent labelling. However, this approach does not provide a direct measurement of cargo escape. Determining what constitutes endosomal escape is therefore highly subjective.

The major disadvantage of both of these fluorescent localisation methods is that they only provide qualitative assessment of endosomal escape (yes/no) but not quantitative information (i.e., they do not measure what proportion of material taken up has escaped). It is challenging to quantify the fluorescence intensity of the cytosolic material, as both cytosolic material and that sequestered in the endosomes/lysosomes will exhibit fluorescence. To overcome this limitation, split green fluorescence protein (GFP) was developed as an endosomal escape probe that allows cytosolic signal to be distinguished from sequestered signal[15–17]. This approach provides a direct quantification of cytosolic delivery, but is limited by poor sensitivity. Typically, concentrations above 10 µM are required to generate a measurable signal or to observe endosomal escape, but these concentrations are significantly higher than therapeutically and clinically relevant doses for most biological materials.

To date, there has not been a direct, highly sensitive and quantitative assay that can distinguish endosomal sequestration from cytoplasmic distribution. There is a significant need for a robust endosomal escape assay that: (i) directly measures cytosolic delivery of the therapeutic; (ii) is highly sensitive so it can detect the low concentrations of material delivered to the cytosol; (iii) is quantitative; and (iv) can determine both the amount of material that escapes and the efficiency of escape.

To address this, we develop a highly sensitive assay for the quantification of endosomal escape based on a split NanoLuciferase reporter system, termed 'Split Luciferase Endosomal Escape Quantification' (SLEEQ) (Fig. 1). The split luciferase assay comprises of two subunits: large BiT protein (LgBiT, 17.8 kDa) and a high affinity complementary peptide (HiBiT, 1.3 kDa)[18]. This assay is useful for measuring intracellular protein interactions (e.g., GPCR homodimerisation)[19], protein dynamics[20] and cellular internalisation (investigating the role of PIP2)[21]. We express LgBiT as a fusion protein with actin, which localises the LgBiT in the cytosol. HiBiT is attached to a protein of interest (GFP) to quantify transport to the cytosol. We demonstrate that SLEEQ can be used to detect picomolar concentrations of proteins delivered to the cytosol, and can quantify the efficiency of endosomal escape. Endosomal escape is a highly inefficient process, with only ~2% of GFP reaching the cytosol in HEK293 cells, and ~7% of GFP reaching the cytosol in HeLa cells. We also apply SLEEQ to explore the endosomal escape efficiency of a range of putative EEPs. While positively charged EEPs increase the total amount of protein delivered to the cytosol, the efficiency of endosomal escape is the same or lower than the efficiency of GFP escape without EEPs. This suggests that the positively charged EEPs increase cytosolic accumulation mostly through non-specific association with the cells, rather than enhancing the efficiency of endosomal escape. Since the EEPs studied here do not increase endosomal escape, a more appropriate name for this group of peptides would be membrane adsorptive peptides.

## Results

**SLEEQ is an ultra-sensitive assay.** Given that endosomal escape is a very inefficient process, it is essential to have an assay that can detect very low concentrations of cytosolic material. Split GFP systems have previously been employed to detect cytosolic delivery of EEPs[16,17,22]. However, the sensitivity of fluorescence techniques is typically limited to micromolar concentrations. To demonstrate the sensitivity of the SLEEQ assay, we compared split NanoLuciferase to split GFP by incubating different concentrations of the small peptide fragment (HiBiT or GFP$_{11}$) with an excess of the larger protein fragment (LgBiT or GFP$_{1–10}$) (Fig. 2a). Our results show a linear correlation with luminescence down to 5 pM of HiBiT for the split NanoLuciferase assay, which is more than four orders of magnitude more sensitive than the detection limit for split GFP (limit of detection = 0.3 µM).

**Development and validation of SLEEQ.** Having demonstrated the superior sensitivity of split NanoLuciferase, we developed SLEEQ to investigate endosomal escape in mammalian cells. First, we engineered cell lines to express LgBiT protein in the cytosol. HEK293 and HeLa cells were transduced with a lentiviral vector encoding the LgBiT transgene and expression of LgBiT was measured using luminescence by lysing the cells and adding HiBiT peptide with the NanoLuciferase substrate, fumarizine. Luminescent

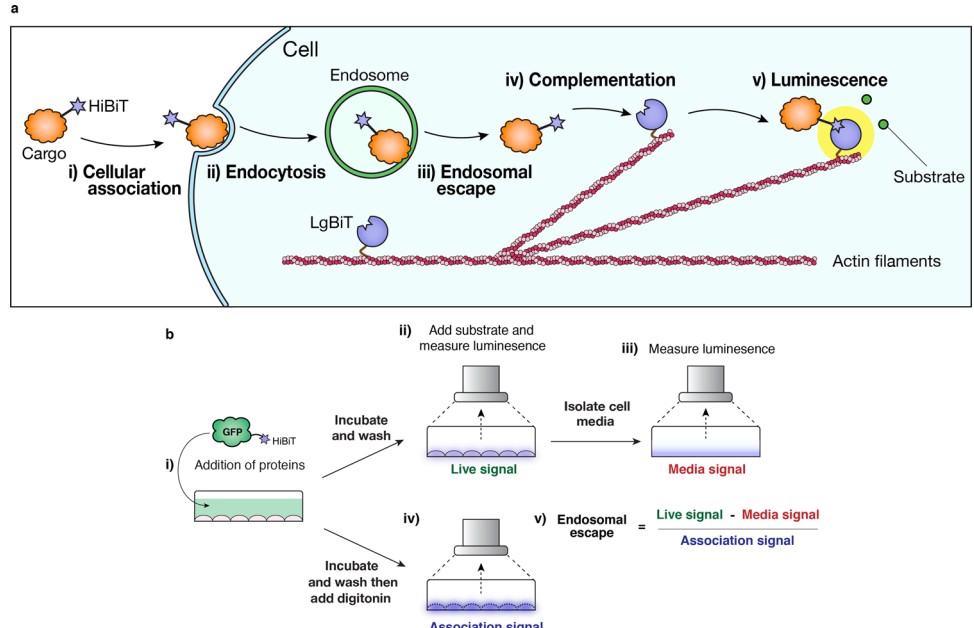

**Fig. 1 The Split Luciferase Endosomal Escape Quantification (SLEEQ) assay enables quantification of endosomal escape in live cells. a** Schematic diagram of how endosomal escape is detected: (i) therapeutic cargo is labelled with HiBiT peptide. (ii) Endocytosis of the HiBiT-tagged cargo results in accumulation in the endosomes. (iii) If endosomal escape occurs, HiBiT-tagged cargo can bind to (iv) LgBiT protein, which is fused to actin filaments to restrict localisation to the cytosol. (v) Complementation between HiBiT and LgBiT forms a functional luciferase enzyme complex, which gives off bright luminescence in the presence of a substrate. **b** Schematic diagram showing how the SLEEQ assay is performed. (i) HiBiT-labelled protein is added to the cells. (ii) After incubation for 4 h, excess protein is removed. Addition of fumarizine substrate enables the luminescence from complemented HiBiT/LgBiT in cytosol to be measured. (iii) To take into account any luminescent signal from excreted LgBiT, the cell media is removed and measured separately. (iv) To determine the total association, digitonin is added to the HiBiT-treated cells to permeabilise cell membranes. This enables any HiBiT trapped in endosomes or stuck to the plasma membrane to complement with LgBiT. (v) Formula for calculating endosomal escape efficiency.

signal in transduced cells ($1.01 \times 10^8$ p/s/cm²/sr) was 732 times higher than in non-transduced cells ($1.38 \times 10^5$ p/s/cm²/sr), indicating successful expression of LgBiT protein (Supplementary Fig. 1). Low background levels of luminescence were observed in transduced cells without added HiBiT ($1.04 \times 10^5$ p/s/cm²/sr). These results demonstrate that both HiBiT peptide and LgBiT protein exhibit minimal background signal on their own. However, we detected 2.7% of the LgBiT protein in the cell media after 24 h (Fig. 2b), which is likely due to secretion of LgBiT protein from the cells. Although secreted LgBiT can be washed from the cells before measuring luminescence, it is possible that the LgBiT/HiBiT complexes could form outside the cells, then be endocytosed by the cell, giving a false estimate of endosomal escape. To limit this, LgBiT was fused to β-actin, a protein found exclusively in the cell cytosol. A SNAP-tag was also incorporated to allow visualisation of LgBiT incorporation into actin filaments by fluorescence microscopy. Clonal cell lines of HEK293 and HeLa cells were established that stably express LgBiT-SNAP-actin (LSA). Cell lysate from the LSA cell lines treated with HiBiT peptide showed comparable luminescence to the free LgBiT (Fig. 2b), suggesting HiBiT still forms an active complex with LgBiT protein when fused to β-actin. Secretion of LSA into the cell media over 24 h was <0.2% of the total expression levels in the stable cell lines, which was ten times lower than secretion of free LgBiT from LgBiT-expressing cells (Fig. 2b), confirming LSA sequestration in the cytoplasm.

Distribution of LgBiT protein throughout the cytosol was confirmed by fluorescence microscopy. HEK293-LSA cells were treated with SNAP-Cell 647-SiR to label the SNAP-tag fused to actin. Figure 2c shows the typical pattern for β-actin staining, with distinct filaments throughout the structure of the cell, and no discernible punctate fluorescence. The concentration of LSA expressed in the cytosol was estimated to be 55.8 nM by

permeabilising the cells with 0.01% w/v digitonin (Supplementary Fig. 2a, c and d). The presence of 0.01% w/v digitonin did not affect the LgBiT luminescence (Supplementary Fig. 2b), and was able to permeabilise endosomes (indicated by the loss of calcein signal in Supplementary Fig. 3). A number of endocytic pathways involve actin; therefore, it is possible that fusing LgBiT to actin could affect protein uptake. To investigate this, we compared the fluid phase endocytosis of wild-type HEK293 and HEK293-LSA cells using calcein, a small impermeable fluorescent dye. There was no significant difference in the uptake of calcein in these two cell lines (Supplementary Fig. 4), suggesting that endocytosis is not significantly affected by the presence of the LgBiT-actin fusion.

**Cationic EEPs increase cytosolic delivery of GFP**. Next, we applied SLEEQ to investigate the endosomal escape of a range of putative EEPs fused to GFP as a model delivery cargo. Each EEP has a different length and amino acid composition; therefore, the total net charge of the proteins varies (Fig. 3c). We chose eight widely used EEPs that have reported endosomal escape capabilities. TAT[23,24], polyarginine (R9)[25,26], 5.3[27] and ZF5.3[27] are cationic arginine containing peptides. The 5.3 peptide incorporates five arginine residues along three helical faces of the avian pancreatic peptide scaffold, whereas ZF5.3 incorporates this arginine topology into a zinc finger domain[27]. ZF5.3 has been shown to induce higher levels of cytosolic delivery than 5.3 when fused to SNAP-tag[28]. We also selected ampiphilic peptides including pHlip[29,30], pHD118[31] and HA2[32]. These peptides are thought to depend on acidification post-internalisation to induce endosomal escape and are commonly referred to as pH-dependent membrane-active peptides. pHlip is derived from bacteriorhodopsin[29,30], whereas pHD118 is a derivitative of bee

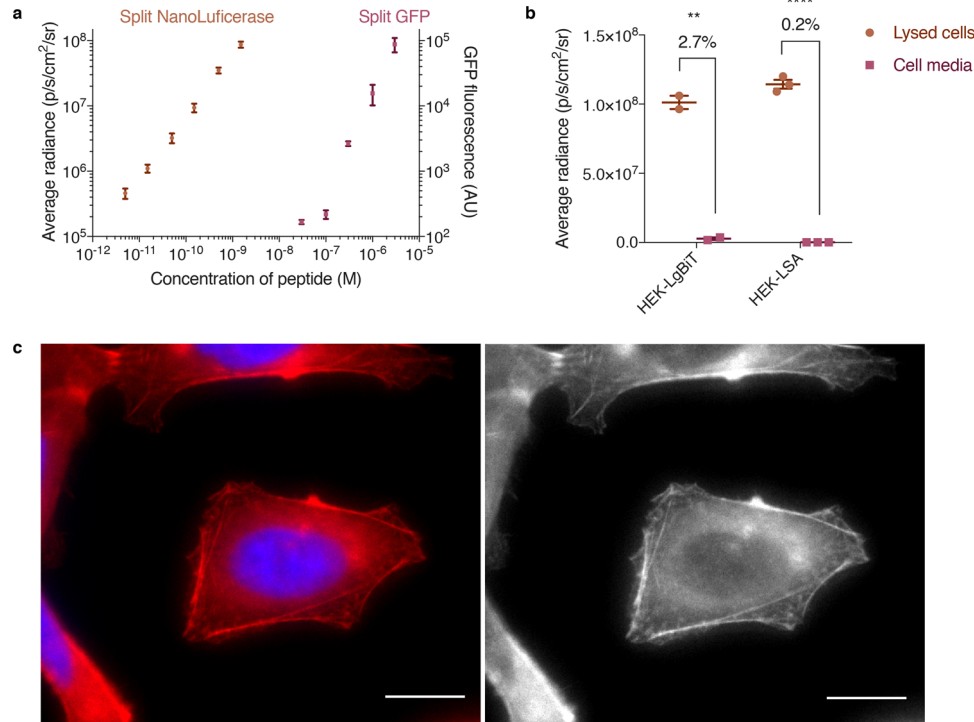

**Fig. 2 SLEEQ is four orders of magnitude more sensitive than split GFP. a** In a 96-well plate assay, the concentration of LgBiT and $GFP_{1-10}$ were fixed at 50 and 3 μM, respectively. LgBiT was sensitive to <5 pM of HiBiT, while $GFP_{1-10}$ required >0.1 μM of $GFP_{11}$ to generate a detectable signal. **b** HEK293 cells with LgBiT expressed in the cytosol (HEK293-LgBiT) excreted 2.7% of the protein into the supernatant. When LgBiT was fused to cytoskeletal protein actin (HEK293-LSA), only 0.2% of the LgBiT was detected in the supernatant. Two-tailed unpaired $t$ test was used to analyse the data. $n = 2$ independent experiments for HEK293-LgBiT and $n = 3$ independent experiments for HEK293-LSA. $**p \leq 0.01$ and $****p \leq 0.0001$. A SNAP-tag was also fused to LgBiT-actin to aid visualisation of the fusion protein. **c** Pseudocoloured (left) and greyscale (right) images of HEK293 cells expressing LSA (red) show cytosolic staining consistent with actin. Nucleus was stained with Hoechst 33342 (blue). $n = 3$ independent experiments Scale bar = 10 μm. Source data are provided as a Source Data file.

venom (melittin)[31]. The influenza haemagglutinin N-terminal peptide HA2 has also been investigated widely, with substitution of certain amino acids with glutamic acid providing improved pH-dependent activity (such as E5)[32,33]. Combination of E5 and TAT (E5TAT peptide) results in a dual arginine-rich and pH-sensitive peptide that ideally employs both arginine-driven cellular association with pH-dependent membrane disruption to increase endosomal escape efficiency[34].

The EEPs were fused to the N-terminus of GFP while HiBiT was fused at the C-terminus. SDS-PAGE shows the purity of the protein yield (Fig. 3a), and the protein size was confirmed by MALDI-TOF mass spectrometry (Supplementary Table 3).

Fusing HiBiT to another protein may affect its luminescent activity. To control for this, the relative activity of each EEP-GFP-HiBiT fusion protein was compared to HiBiT peptide by combining the fusion proteins with an excess of purified recombinant LgBiT protein. The luminescent signal for the fusion proteins was between 20 and 80% of the free HiBiT peptide (Fig. 3b). Given the sensitivity of HiBiT peptide is ~5 pM, we determined that the loss in activity would not significantly affect the sensitivity of the assay.

Next, we investigated the cytosolic accumulation of these fusion proteins in HEK293 cells. After 4 h of incubation with 1 μM EEP-GFP-HiBiT, the cells were washed to remove unbound material and luminescence was measured. The EEP-GFP-HiBiT proteins exhibited minimal cell toxicity at this concentration, and the cells remained viable throughout the experiment (Supplementary Figs 5 and 6). Signal in the cell supernatant was also sampled to correct for any cell death that could contribute to false positive signals. To account for the differences in HiBiT activity

for each EEP-GFP-HiBiT protein (Fig. 3b), the cytosolic and total cellular signal was normalised to the activity of free HiBiT (Fig. 4a, b). GFP without an EEP served as a baseline for endosomal escape, as any cytosolic delivery of GFP should be due to constitutive levels of cytosolic transport.

R9, TAT, ZF5.3 and E5TAT showed significantly enhanced delivery (7- to 30-fold increase) of GFP to the cytosol (Fig. 4a). Surprisingly, cytosolic delivery of 5.3, pHD118, pHlip and HA2 was not significantly higher than GFP alone (Fig. 4a). As expected, GFP showed low delivery to the cytosol. A similar trend was observed in HeLa cells (Supplementary Fig. 7a).

Western blot analysis of cell lysate shows that GFP protein remains intact. No band corresponding to the molecular weight of HiBiT peptide was observed (Supplementary Fig. 8).

**Cationic EEPs enhance total cellular association.** While R9, TAT, ZF5.3 and E5TAT peptides significantly increased cytosolic accumulation compared to GFP without an EEP, all these EEPs have a positive charge. It is well established that positively charged proteins associated strongly with negatively charged plasma membranes[35]. Therefore, to decouple cytosolic accumulation from the total amount of protein adsorbed to the cell, we determined the total cellular association (both surface-bound and internalised material) of the proteins. To do this, cells were washed stringently after incubation to remove all unbound materials and treated with digitonin for 1 h to permeabilise all cellular membranes (Fig. 4b).

As expected, the positively charged EEPs (R9, TAT, ZF5.3, E5TAT and 5.3) showed significantly higher cellular association (29- to 49-fold) than GFP (Fig. 4b). In HeLa cells, a similar trend

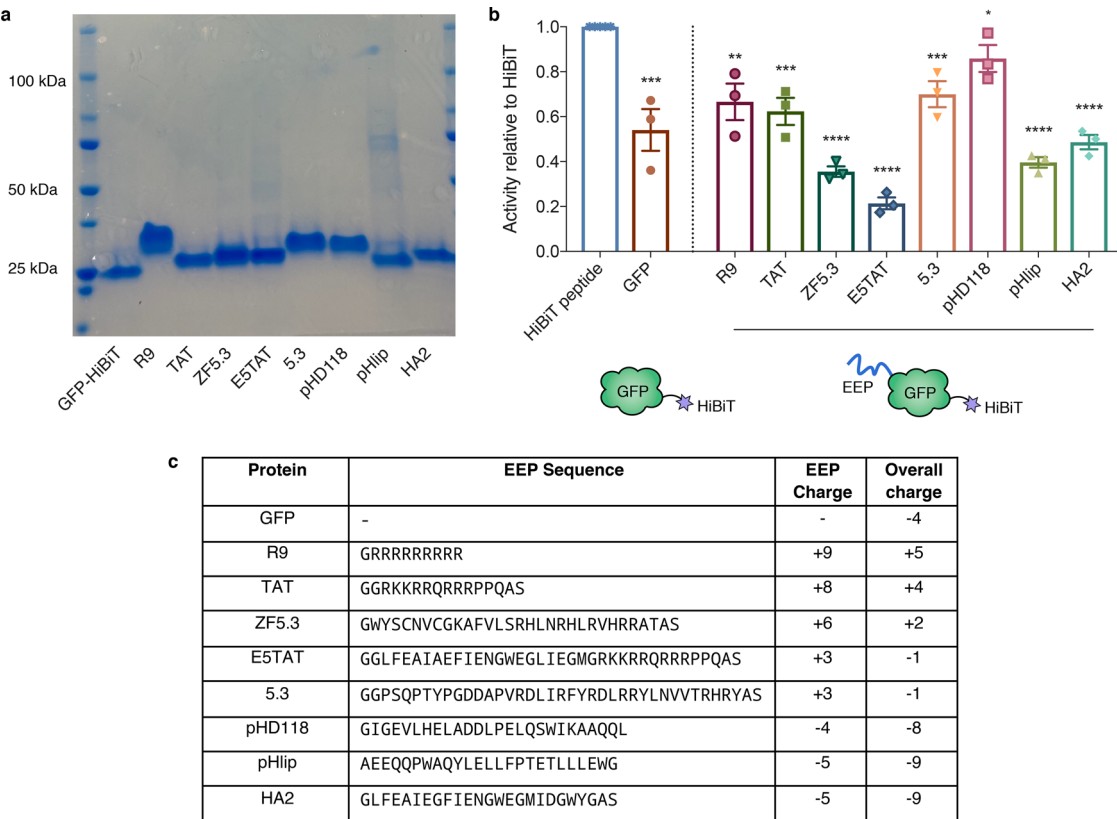

**Fig. 3 HiBiT fused to EEP-modified GFP retains luminescent activity. a** SDS-PAGE gel of EEP-GFP-HiBiT fusion proteins. $n = 3$ independent experiments **b** Luminescent activity of EEP-GFP-HiBiT relative to HiBiT peptide (activity of HiBiT normalised to 1). For simplicity in naming, EEP-GFP-HiBiT fusion proteins are labelled by the EEP. Data represents mean ± SEM, $n = 3$ independent experiments. Two-tailed unpaired $t$ test was used to analyse the data. *$p \leq 0.05$, **$p \leq 0.01$, ***$p \leq 0.001$ and ****$p \leq 0.0001$. **c** Summary of EEP peptide sequences, their respective charges and the overall charge when fused to GFP-HiBiT. EEPs are listed in order of decreasing charge. Source data are provided as a Source Data file.

with the total cellular association was observed (30- to 79-fold, Supplementary Fig. 7b). Negatively charged EEPs (pHD118, pHlip and HA2) showed similar association to GFP in both cell lines. This highlights how positively charged EEPs can significantly influence interaction with the plasma membrane.

To further investigate the intracellular distribution of EEP-GFP-HiBiT, the cells were imaged using confocal microscopy (Fig. 5). As predicted by the cellular association results (Fig. 4b), the fluorescence signal from the negatively charged proteins was significantly lower than the signal from the positively charged proteins. To examine the distribution of EEP-GFP-HiBiT within the cells, the dynamic range of each image was optimised to aid visualisation. Unadjusted images for comparison of signal intensity can be found in Supplementary Fig. 9. All proteins showed distinct, punctate fluorescence, indicating entrapment within endosomal/lysosomal compartments and minimal endosomal escape. E5TAT also displayed pronounced membrane association on the cell surface.

To determine the endosomal escape efficiency of the different EEPs, we calculated the ratio of the cytosolic signal divided by the total associated signal (Fig. 4c). The endosomal escape efficiency of GFP alone was approximately <2% in HEK293 cells, which was lower than that of HeLa cells (~7%, Supplementary Fig. 7c, d). Strikingly, none of the EEPs tested had higher endosomal escape efficiency than GFP alone. Even more significantly, while several of the positively charged EEPs showed higher total cytosolic delivery, the endosomal escape efficiency was significantly lower than GFP alone. Overall, these results suggest that none of the EEPs tested were more efficient at delivering protein to the cytosol than the apparent constitutive cytosolic transport mechanisms.

While low endosomal escape efficiency was observed for all EEPs, it is possible that the high cell association could be saturating the endosomal escape capacity of the cells. To probe this, we attempted to match total cellular association of the proteins by adjusting the concentrations of protein added to the cells (Fig. 6). Free HiBiT peptide was also included in these experiments to determine if HiBiT has any endosomal escape properties.

When the concentration of protein was adjusted to give similar total cellular association, the positively charged EEPs no longer exhibited increased cytosolic accumulation. In fact, these EEPs exhibited the same or significantly less cytosolic delivery than GFP. This translates to a significantly lower percentage of endosomal escape compared to GFP without an EEP (Fig. 6c, d), which is consistent with the endosomal escape percentages shown in Fig. 4d when cells were treated with equal concentrations of GFP. HiBiT peptide alone showed the lowest endosomal escape efficiency (Fig. 6c, d), indicating that HiBiT does not possess endosomal escape properties. Similar results were observed for HeLa cells (Supplementary Fig. 10).

It has been suggested that higher concentrations of EEPs may be required to induce endosomal escape. No significant increase in endosomal escape efficiency was observed when HEK293 cells were treated with 10 μM of EEP-GFP-HiBiT (Supplementary Fig. 11). It has also been proposed that the presence of lipopolysaccharides (LPS) from the bacteria used to express the EEP-GFP-HiBiT could interfere with endosomal escape. To control for this, we performed the SLEEQ assay using protein synthesised in ClearColi[36], a strain of *Escherichia coli* that does not contain LPS. We observed no difference in the endosomal escape efficiency of GFP-HiBiT and TAT-GFP-HiBiT synthesised

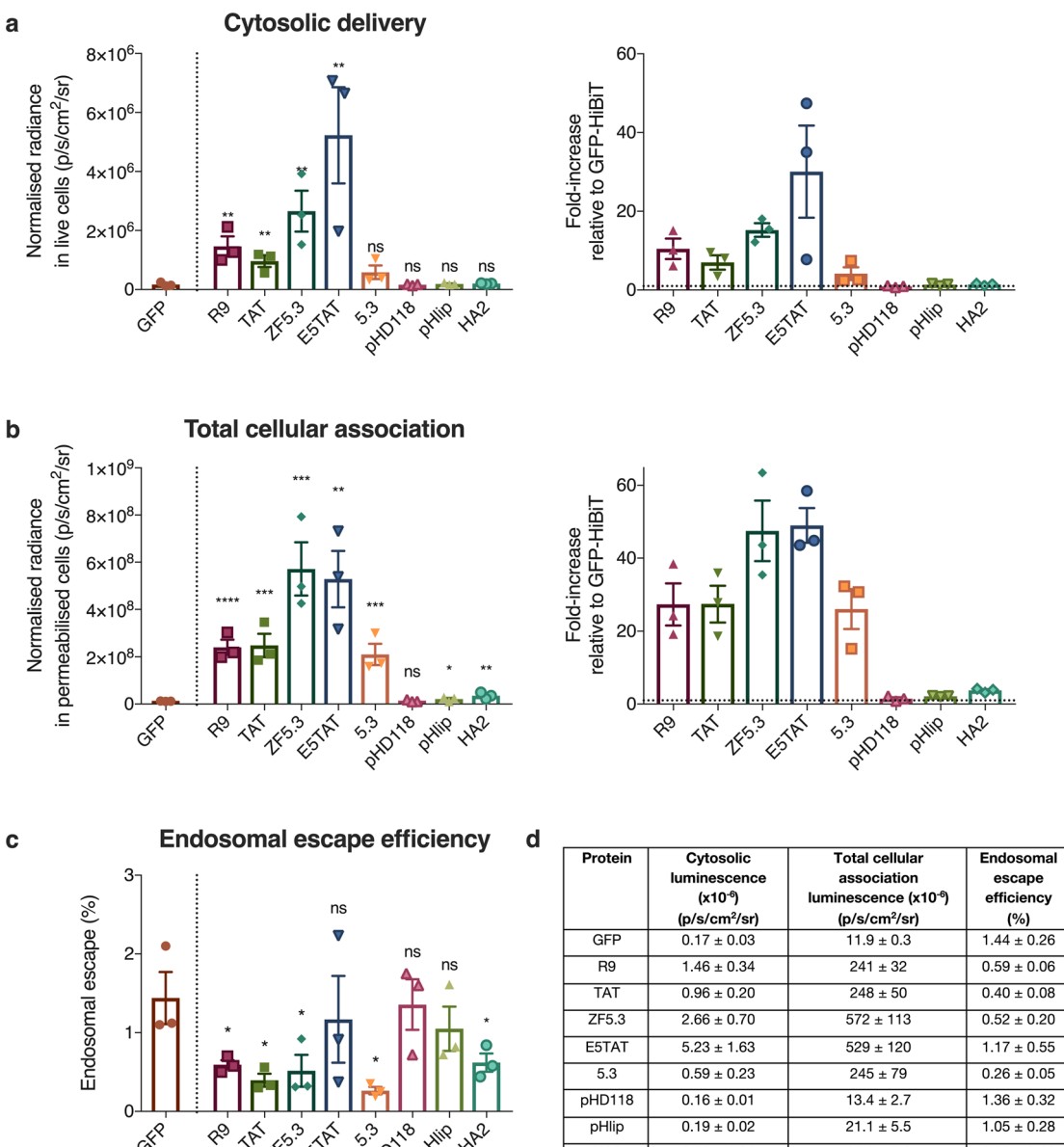

**Fig. 4 Cationic EEPs increase cytosolic delivery of GFP but do not increase endosomal escape efficiency. a** Cytosolic luminescent signal of EEP-GFP-HiBiT in HEK293-LSA cells and fold-increase in signal with respect to GFP (represented by dotted line = 1). HEK293-LSA cells were incubated with EEP-GFP-HiBiT proteins at 1 μM for 4 h. Data represents mean ± SEM, $n = 3$ independent experiments. Kruskal–Wallis with uncorrected Dunn's test was used to analyse the data. **b** Total cellular association of EEP-GFP-HiBiT in HEK293-LSA cells and fold-increase with respect to GFP (represented by dotted line = 1) determined by permeabilising the cells using 0.01% w/v digitonin. Data represents mean ± SEM, $n = 3$ independent experiments. Kruskal–Wallis with uncorrected Dunn's test was used to analyse the data. **c** Endosomal escape efficiency of EEP-GFP-HiBiT proteins determined by ratioing cytosolic signal with total cellular association. Data represents mean ± SEM, $n = 3$ independent experiments. Kruskal–Wallis with uncorrected Dunn's test was used to analyse the data. **d** Summary of cytosolic luminescence, total cellular association luminescence and endosomal escape efficiency for all proteins. Data represents mean ± SEM, $n = 3$ independent experiments. Kruskal–Wallis with uncorrected Dunn's test was used to analyse the data, with GFP being the control group. ns (not significant) denotes $p > 0.05$, $*p \leq 0.05$, $**p \leq 0.01$, $***p \leq 0.001$ and $****p \leq 0.0001$. Source data are provided as a Source Data file.

in ClearColi or *E. coli* (Supplementary Fig. 12). Taken together, these results suggest that EEPs do not improve endosomal escape efficiency, but rather just increase cell association.

## Discussion

The process of endosomal escape is postulated to be highly inefficient. However until now, our estimates of endosomal escape efficiency have been qualitative. Image-based techniques to measure endosomal escape are inherently subjective and

qualitative. The majority of material delivered to the cell is concentrated within endosomes and lysosome, as highlighted by the punctate fluorescence observed in Fig. 5. The signal from these compartments can easily swamp the diffuse signal from material that has escaped from the endosome. Furthermore, the signal from out of focus endosomal compartments is often confused with endosomal escape. The images in Fig. 5 and Supplementary Fig. S6 highlight the limitations of using fluorescence microscopy to investigate endosomal escape. It is challenging to compare images with such highly contrasting intensities, and it is also

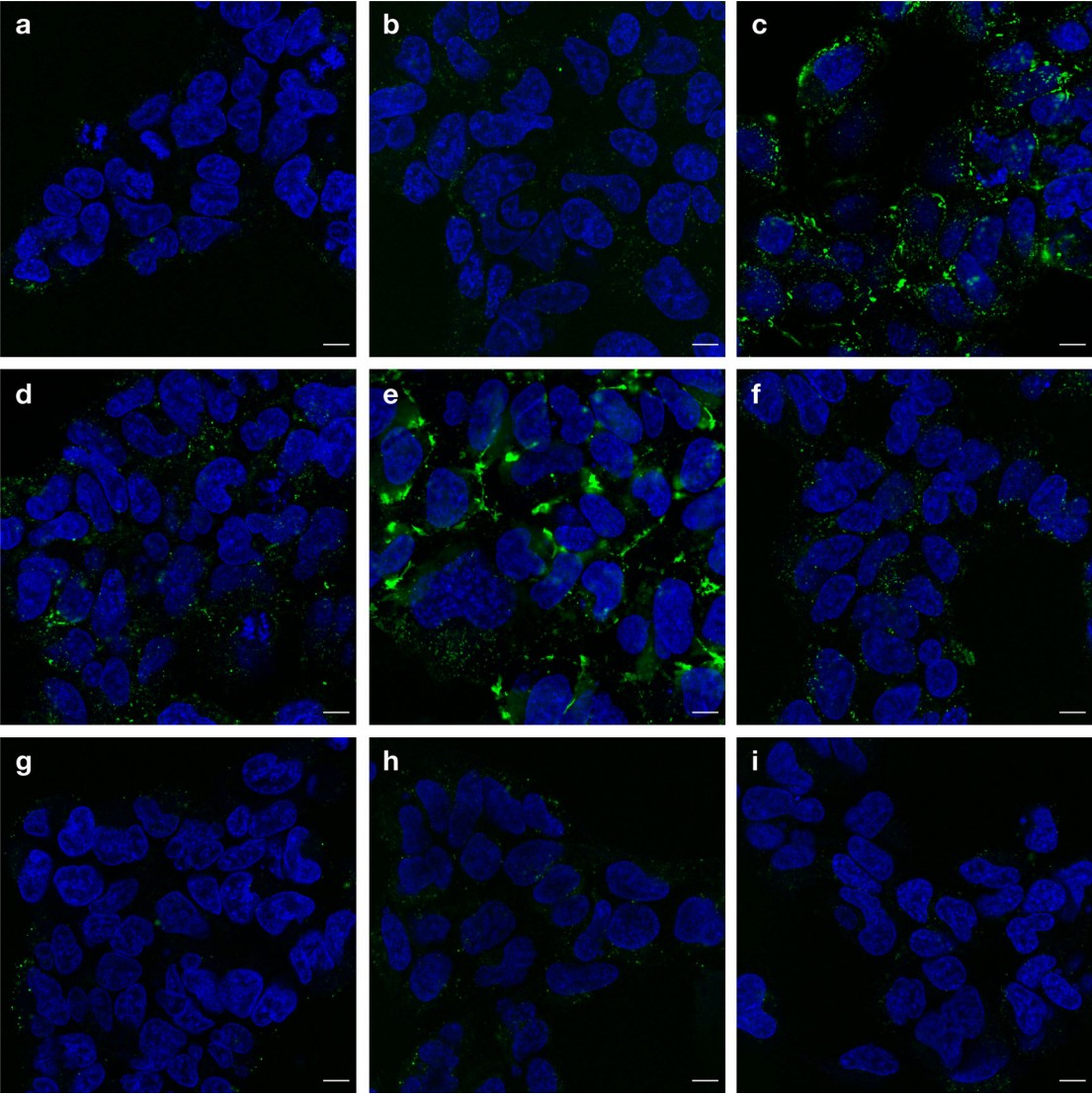

**Fig. 5 EEP-GFP-HiBiT fusion proteins exhibit punctate staining, suggesting limited endosomal escape.** HEK293-LSA cells treated with 1 µM EEP-GFP-HiBiT (green) proteins: **a** GFP, **b** R9, **c** TAT, **d** ZF5.3, **e** E5TAT, **f** 5.3, **g** pHD118, **h** pHlip and **i** HA2 at 1 µM for 4 h. Cells were washed and nuclei were stained with Hoechst 33342 (pseudocoloured blue) before confocal microscopy. $n = 2$ independent experiments Scale bar = 10 µm.

challenging to observe weak cytosolic signal, as the bright punctate fluorescence in the endosomes overwhelms any diffuse cytosolic signal. To overcome this, techniques such as the chloroalkane penetration assay[37], fluorescence correlation spectroscopy (FCS)[28] and NanoClick[38] have been developed and while they provide a way of measuring the amount of material that has reached the cytosol, they do not measure the efficiency of endosomal escape. Furthermore, FCS is also limited by a need to manually select a point in the cell to measure the cytosolic concentration of fluorescently labelled protein. As the cytosol is a highly dynamic environment, it can be challenging to ensure that this point solely measures cytosolic material and excludes material from endosomal compartments that may traffic through the point during the analysis period.

To improve the quantification endosomal escape, sensors that signal when cargo reaches the cytosol are required. The first generation of 'switch on' endosomal escape sensors were based on a split GFP[16,17,22]. This sensor significantly improves the ability to detect endosomal escape. However, as with fluorescence microscopy, the limited sensitivity of fluorescence techniques means the limit of detection is high and concentration of cargo

delivered to the cells needs to be much higher than the therapeutically relevant doses. This means that endosomal escape within a therapeutically relevant window cannot be studied. A much more sensitive probe based on inactivated Renilla luciferase (RLuc) protein that is activated upon deglycosylation in the cytosol has recently been reported[39,40]. This method enables correlation of endosomal escape with transfection efficiency of the delivered mRNA, but does not quantify endosomal escape efficiency, and employs a large glycosylated protein (~75 kDa) as the sensor. Here, we have demonstrated that SLEEQ is a robust and highly sensitive assay that allows direct quantification of endosomal escape using a short peptide sensor. The assay is greater than four orders of magnitude more sensitive than a split GFP system (Fig. 2). The limit of detection for the SLEEQ assay is $2.1 \times 10^4$ p/s/cm$^2$/sr, which is >8 times lower that the signal detected for GFP (GFP = $1.89 \times 10^5$ p/s/cm$^2$/sr). Western blot analysis of cell lysate (Supplementary Fig. 8a) shows that the fusion protein largely remains intact and no degradation of the GFP-HiBiT fusion was detected over the period of the experiment (Supplementary Fig. 8b). More importantly, no low molecular weight band corresponding to free HiBiT peptide was observed,

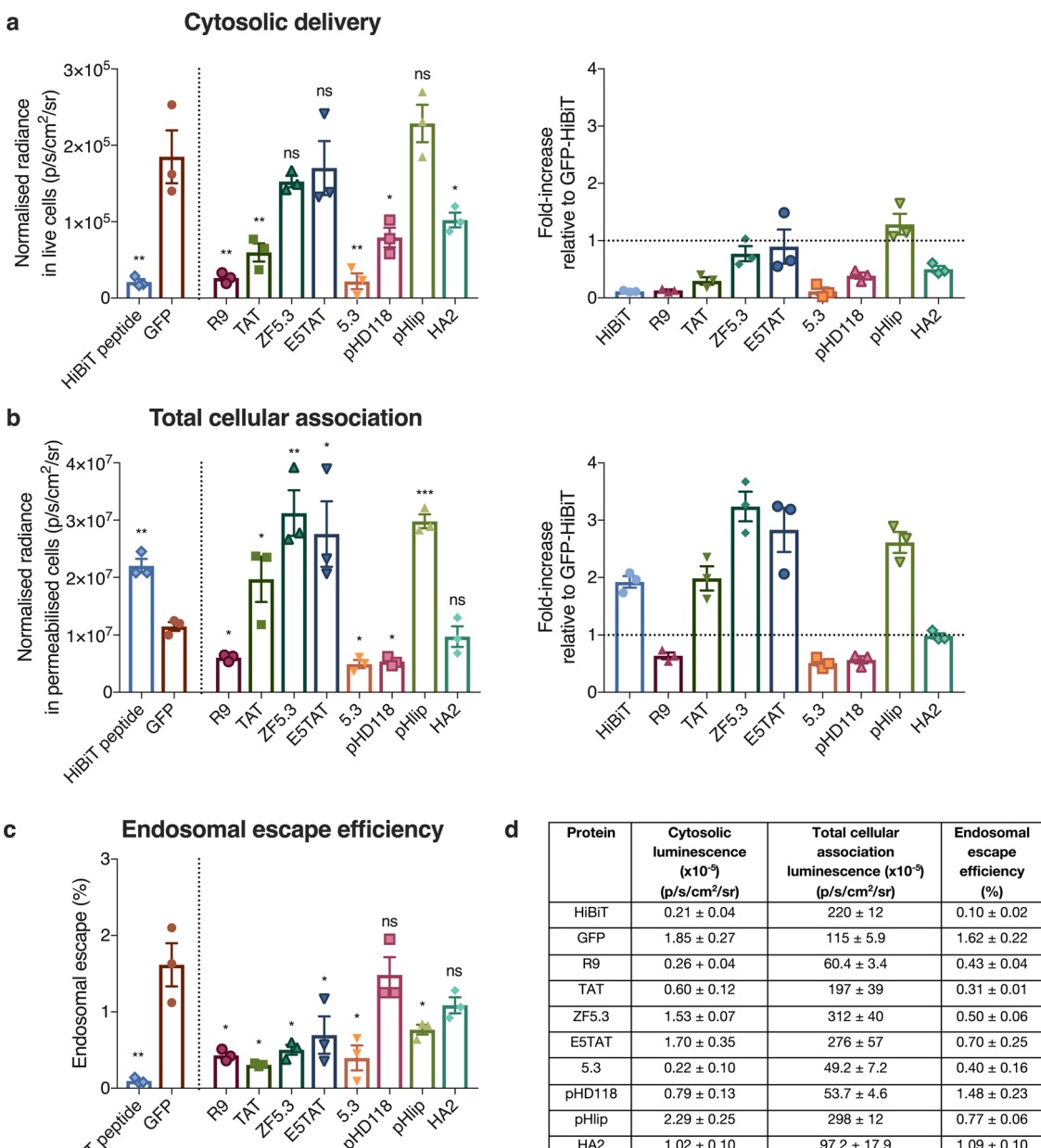

**Fig. 6 EEPs do not increase endosomal escape efficiency of GFP. a** Cytosolic luminescent signal of EEP-GFP-HiBiT in HEK293-LSA cells and fold-increase in signal with respect to GFP (represented by dotted line = 1). HEK293-LSA cells were incubated with EEP-GFP-HiBiT proteins at varying concentrations (refer to methods) for 4 h. Data represents mean ± SEM, $n = 3$ independent experiments. Kruskal–Wallis with uncorrected Dunn's test was used to analyse the data. **b** Total cellular association of EEP-GFP-HiBiT in HEK293 cells and fold-increase with respect to GFP-HiBiT (represented by dotted line = 1) determined by permeabilising the cells using 0.01% w/v digitonin. Data represents mean ± SEM, $n = 3$ independent experiments. Kruskal–Wallis with uncorrected Dunn's test was used to analyse the data. **c** Endosomal escape efficiency of EEP-GFP-HiBiT proteins determined by ratioing cytosolic signal with total cellular association. Data represents mean ± SEM, $n = 3$ independent experiments. Kruskal–Wallis with uncorrected Dunn's test was used to analyse the data. **d** Summary of cytosolic luminescence, total cellular association luminescence and endosomal escape efficiency for all proteins. Data represents mean ± SEM, $n = 3$ independent experiments. Kruskal–Wallis with uncorrected Dunn's test was used to analyse the data, with GFP as the control group. Data represents mean ± SEM, $n = 3$ independent experiments. ns (not significant) denotes $p > 0.05$, $*p ≤ 0.05$, $**p ≤ 0.01$ and $***p ≤ 0.001$. Source data are provided as a Source Data file.

indicating that the luminescent signal measured in the SLEEQ assay comes from intact fusion protein, and not degradation products of the protein.

Our results support the contention that endosomal escape is an inefficient process. Only <2% of GFP associated with HEK293 cells was able to gain entry into the cytosol. This demonstrates not only the inefficiency of endosomal escape, but importantly highlights the sensitivity of SLEEQ to be able to

quantify such low levels of endosomal escape. We also investigated endosomal escape in HeLa cells, and found they exhibited significantly higher endogenous levels of endosomal escape (~7%) than the HEK293 cells. This suggests that HeLa cells may have naturally leakier endosomes than HEK293 cells. These results demonstrate differences in intracellular trafficking in different cell lines and highlight the importance of studying different cell types.

The most notable result from the SLEEQ assay is that none of the putative EEPs increased the endosomal escape efficiency of the protein they were fused to. When dosed at 1 µM, positively charged peptides (R9, TAT, ZF5.3 and E5TAT) enhanced cytosolic delivery of GFP-HiBiT in both HEK293 (7- to 30-fold increase) and HeLa (7- to 21-fold increase) cells (Fig. 4a and Supplementary Fig. 7a). Although 5.3 peptide is also positively charged, it did not appear to significantly accumulate in the cytosol of HEK293 cells, but showed significant cytosolic delivery in HeLa cells. The positively charged peptides increased association with cells by an even greater amount (Fig. 4b and Supplementary Fig. 7b). When the concentration of the proteins was lowered to match the association of GFP, the positively charged peptides showed significantly lower cytosolic delivery than unmodified GFP. This demonstrates two important points: (1) the increased cytosolic delivery of cationic EEPs seen at 1 µM concentration (Fig. 4a) can be attributed to increased accumulation with the cells, likely driven by electrostatic interactions with the negatively charged plasma membrane. (2) The positive charge of the EEPs actually hinders the cytosolic delivery when similar amounts of protein are associated with the cells (Fig. 6a). It is likely that the electrostatic interaction with the EEPs and the negatively charged membrane is retained in the endosome and they are unable to dissociate from the endosomal membranes, which inhibits the endogenous leakage of proteins into the cytosol. Negatively charged EEPs (pHD118, pHlip and HA2) showed no increase in cytosolic delivery and no significant increase in their total cellular association.

There is a significant body of evidence in the literature[7,41], and backed up by our microscopy images (Fig. 5), to show that the majority of material delivered to cells using EEPs is endocytosed and becomes trapped inside endosomes. If cells are incubated with the proteins at 4 °C to inhibit endocytosis, there is a significant drop in the cytosolic signal detected (similar to the background luminescence—Supplementary Fig. 13). Treatment of cells with chemical inhibitors of endocytosis (EIPA, NaN₃, 2-deoxy-D-glucose, cytochalasin D and wortmannin) resulted in significantly elevated levels of LgBiT in the media (Supplementary Fig. 14), indicating that stress induced by these treatments compromise membrane integrity, which could lead to unreliable measurements of cytosolic delivery. These results highlight the limitations of using pharmacological inhibitors to probe complex phenomena such as cytosolic transport. It cannot be ruled out that at 37 °C a small amount of protein reaches the cytosol as a result of translocation across the plasma membrane; however, there is no direct evidence to show this occurs, nor is there evidence to suggest that the plasma membrane is more permeable than the endosomal membrane. Importantly, the results here show that regardless of if cytosolic delivery is the result of endosomal escape or direct plasma membrane translocation, the presence of the EEP does not increase the efficiency of delivery to the cytosol.

These results fundamentally change our understanding of the role of EEPs and how they promote cytosolic delivery. Rather than improving endosomal escape efficiency, it appears that when positively charged EEPs are fused to a protein, they increase cytosolic delivery simply by increasing cellular accumulation. These experiments focus on EEPs fused to a cargo and do not exclude the possibility that EEPs delivered separately from the cargo protein (i.e., not as a fusion protein but as a separate endosomal escape agent) or when fused to a short peptide, could promote endosomal escape. However, delivery of an EEP separately from the intended therapeutic would induce non-specific endosomal escape in a range of cells, which could lead to significant side effects. Therefore, it is highly desirable for EEPs to be functional as a fusion protein. The increased cytosolic delivery of the EEP fusion proteins comes at the cost of reduced endosomal escape efficiency, as the majority of protein appears to remain electrostatically bound to the membrane.

In summary, SLEEQ is a highly sensitive, direct and quantitative method for detecting endosomal escape. We demonstrated that endosomal escape is an inefficient process that varies between cell lines. In an effort to enhance endosomal escape, we tested a range of EEPs that have been widely studied to enhance cytosolic delivery. Our findings demonstrate that while positively charged EEPs fused to a model protein improved cytosolic accumulation, it was not via enhancement of endosomal escape. Increased cytosolic accumulation was a result of an increase in total association of EEPs with cells. Unlike existing assays, SLEEQ enables the detection of constitutive levels of cytosolic transport and is a powerful assay that has the potential to quantify endosomal escape of a range of biological therapies at therapeutically relevant concentrations.

## Methods

**Plasmid construction**. All plasmids were constructed using NEBuilder HiFi DNA assembly master mix (NEB) with PCR products, vector restriction digests or DNA oligonucleotides with compatible overhangs. Cloning was performed in TOP10 chemically competent *E. coli* (Thermo Fisher Scientific).

GFP (muGFP)[42] with a C-terminal HiBit peptide (NlucC variant #86: VSGWRLFKKIS)[18] was fused to the C-terminus of 14x His bdSUMO[43] and inserted into pET His6 MBP TEV LIC cloning vector (2M-T), a gift from Scott Gradia (RRID:Addgene_29708). The vector's 6x His MBP TEV coding sequence was replaced with a 14x His bdSUMO–E5-TAT–GFP–HiBiT gene fragment purchased from Integrated DNA Technologies (IDT). EEP sequences were purchased as DNA oligonucleotides from IDT and subcloned to plasmids not used in this study, before being PCR amplified with overhangs for assembly with a PCR product of the pET 14x His bdSUMO vector (above). An additional glycine was inserted before R9 to improve bdSUMO cleavage.

pSF1389 encoding bdSENP1 was a gift from Dirk Görlich (RRID: Addgene_104962). The sequence encoding GFP₁₋₁₀ for the split GFP assay[44] was ordered as a gene fragment from IDT and was also inserted into pET His6 MBP TEV LIC cloning vector (2M-T) where the MBP sequence was replaced with GFP₁₋₁₀.

The LSA lentiviral plasmid was constructed by insertion of LgBiT DNA[18], SNAP-tag DNA (pSNAPf, New England Biolabs) and β-Actin DNA (Actin mRFP-PAGFP was a gift from Guillaume Charras & Tim Mitchison, RRID: Addgene_62382) into the third-generation lentiviral plasmid pCDH-EF1-IRES-Puro (System Biosciences).

A complete list of primers used can be found in Supplementary Table 1. The constructed plasmid pCDH-EF1-LgBiT-SNAP-actin-IRES-puro (Plasmid #139103) is available from Addgene.

**Cell culture**. Dulbecco's Modified Eagle Medium (DMEM) (4.5 g/L glucose, 110 mg/L sodium pyruvate, no glutamine), 100x GlutaMAX supplement, Dulbecco's phosphate-buffered saline (no calcium, no magnesium) and TrypLE were purchased from Thermo Fisher Scientific.

Human embryonic kidney cells (HEK293, ATCC Cat# CRL-1573, RRID: CVCL_0045) and human cervical cancer epithelial cells (HeLa, a gift from David Jans) were cultured in DMEM supplemented with 10% fetal bovine serum (FBS), 1x GlutaMAX and 100 units/mL streptomycin and 100 µg/mL penicillin. Both cell lines were maintained at 37 °C in a 5% CO₂ atmosphere, and sub-cultured every 2–4 days when 70–90% confluency was reached.

HEK293 and HeLa cells that stably expresses LSA were maintained in DMEM supplemented with 10% FBS, 1x GlutaMAX and 100 units/mL streptomycin, 100 µg/mL penicillin and 2 µg/mL puromycin every five passages. Only cells with less than 30 passages were used in all experiments. Cells were tested negative for mycoplasma contamination.

**Generation of stable cell lines expressing LgBiT-SNAP-actin**. Lentivirus was produced by transfecting HEK293-FT cells (Thermo Fisher Scientific) with the third-generation lentiviral vector system using lipofectamine 3000 (Thermo Fisher Scientific). Lentivirus was harvested 48 h after transfection and applied to relevant cell lines. Cells transduced with lentivirus were grown until confluent then selected with 2 µg/mL puromycin for positive incorporation of the transfer gene. Resistant cells were then single cell sorted into 96-well plates using a MoFlo Astrios (Beckman Coulter) to begin clonal cell lines.

**Expression of LgBiT in cells**. To determine whether LgBiT protein was successful expressed in cells, non-transduced HEK293 cells and HEK293-LgBiT cells were seeded at 10,000 cells/well in black 96-well clear bottom plates. After overnight

incubation, cells were treated with 1 nM HiBiT peptide and 0.01% wt/v digitonin for 1 h at 37 °C to completely permeabilise the cells. NanoGlo Live Cell substrate was added to cells and luminescence was measured on In Vivo Imaging System Lumina (IVIS) Lumina II.

**Sensitivity of split NanoLuciferase and split GFP.** The sensitivity of split NanoLuciferase and split GFP was assessed by combining a fixed amount of purified LgBiT or GFP$_{1-10}$ protein with increasing molar equivalents of HiBiT or GFP$_{11}$ peptide. For split NanoLuciferase, 25 μL of desired concentrations (half-log increases) of HiBiT peptide were combined with 50 μL of 100 nM LgBiT protein in a black 96-well clear bottom microplate. To the mixture, 25 μL of NanoGlo Live Cell Substrate (Promega) was added and luminescence was measured on the IVIS Lumina II (Perkin Elmer) 10 min after substrate addition. Luminescence data were processed with Living Image 4.3.1 software. Luminescence was quantified in average radiance in units of photons per second per centimetre squared per steradian (p/s/cm$^2$/sr).

For split GFP, 50 μL of desired concentrations of GFP$_{11}$ peptide (GL BioChem) was combined with 50 μL of 6 μM GFP$_{1-10}$ for a final volume of 100 μL in a black 96-well clear bottom microplate. Incubation proceeded at 37 °C inside ClarioSTAR microplate reader (BMG Labtech) with fluorescence emission at 515 nm being detected after 470 nm excitation 2.75 h post GFP$_{11}$ addition (the time at which peak fluorescence intensity was observed).

Radiance and fluorescence values recorded were averages of three experiments subtracted by the average radiance or fluorescence values of blank media or PBS.

**Secretion of LgBiT.** To determine LgBiT secretion, cells expressing LgBiT only or LSA were seeded at 10,000 cells/well to a final volume of 100 μL in black 96-well clear bottom microplate and incubated at 37 °C for 24 h. Then, 50 μL of the cell supernatant was removed carefully without touching the cells adhered to the bottom of the plate and transferred to another black 96-well clear bottom microplate. The remaining volume of media was discarded and the cells were washed once with media.

In all, 50 μL of 0.02% w/v digitonin and 25 μL of 4 nM HiBiT peptide (diluted in DMEM supplemented with 10% FBS) were added to cells. Then, 25 μL of 4 nM HiBiT peptide was added to the collected cell supernatant. Both cells and cell media were incubated for 1 h at 37 °C and 25 μL of NanoGlo Live Cell Substrate was added to both cell media and cells and luminescence was measured on IVIS Lumina II 10 min after substrate addition. Exposure time was set to 10 s. Cells and cell supernatant that were not treated with HiBiT peptide were blank samples. Percentage of LgBiT secretion from cells was calculated according to the equation below:

$$\text{Percentage of LgBiT secretion}(\%) = \frac{\text{Average radiance(supernatant)} - \text{average radiance(blank)}}{\text{Average radiance(cells)} - \text{average radiance(blank)}} \times 100\% \quad (1)$$

**Digitonin permeabilisation.** To determine the concentration of digitonin required for total cell permeabilisation, HEK cells expressing LSA were seeded 1 day prior at 10,000 cells/well. Cells were incubated with HiBiT peptide at 1 μM for 2 h. Cells were then washed thrice with cell growth media (DMEM supplemented with 10% FBS). Digitonin stock solution (prepared in DMSO as 10% w/v) was diluted and added to cells for final concentrations of 0.001, 0.005, 0.01, 0.02 and 0.05% w/v for 30 min at 37 °C. NanoGlo Live Cell Substrate, prepared according to the manufacturer's instructions, was added to cells and luminescence was measured on the IVIS 30 min after substrate addition. Exposure time was set to 10 s.

To further confirm this, a calcein assay was also performed. Cells were seeded 1 day prior at 10,000 cells/well. Cells were incubated with 100 μg/mL calcein for 2 h, and then washed thrice with cell media. Cells were then treated with 0.004, 0.01, 0.05% w/v digitonin and stained with Hoechst 33342 and propidium iodide (PI). Imaging was performed using Olympus IX83 deconvolution wide field microscope with a 60x silicone objective. All images were processed using SlideBook 6.0 and ImageJ Fiji 2.0.0 software.

To determine whether digitonin affects luciferase activity, purified LgBiT (50 nM) was combined with various concentrations of HiBiT peptide (0.03, 0.1, 0.3, 1, 3, 10 nM) with or without the presence of 0.01% w/v digitonin. No difference in emitted radiance was observed with or without digitonin.

**Protein expression.** Plasmids were freshly transformed into BL21 Star (DE3) *E. coli* (Thermo Fisher Scientific) or ClearColi BL21 (DE3) (Lucigen) prior to each expression batch. Transformed bacteria were directly inoculated into 2 L plastic baffled flasks (Thomson Instrument Company) containing 200 mL optimised growth medium with 15 g/L tryptone (Thermo Fisher Scientific), 30 g/L yeast extract (Thermo Fisher Scientific), 8 mL/L glycerol (Promega), 10 g/L NaCl and shaken at 200 RPM overnight at 37 °C. High-density cultures were then reduced to room temperature and induced with 0.4 mM IPTG (Roche) for 6 h. Bacteria were harvested by centrifugation at 4000 *g*. Bacterial pellets were stored at −20 °C if processing was not immediate. The full sequences of each protein can be found in Supplementary Fig. 15.

**Protein purification.** Bacterial pellets were resuspended in a high salt buffer (1 M NaCl, 50 mM Imidazole, 50 mM monosodium phosphate, adjusted to pH 8.0) supplemented with complete EDTA-free protease inhibitors, 2 mM MgCl$_2$ and benzonase. Resuspended bacteria were lysed by homogenisation with an EmulsiFlex-C3 (Avestin) before centrifugation at 12,000 *g* and clarified through a 0.45-μm syringe filter to remove cellular debris. Protein was purified by immobilised metal affinity chromatography (IMAC) using Protino Ni-NTA agarose (Machery-Nagel). Captured protein was washed copiously with high salt buffer and a low salt buffer (100 mM NaCl, 50 mM Imidazole, 50 mM monosodium phosphate, adjusted to pH 8.0) before elution (300 mM NaCl, 450 mM Imidazole, 50 mM monosodium phosphate, adjusted to pH 8.0). Eluted GFP fusion proteins were concentrated and buffer exchanged into buffer PB (500 mM NaCl, 50 mM Tris-HCl, pH 8.0) using 30 kDa Amicon centrifugal filters (Merck). Eluted bdSENP1, LgBiT and GFP$_{1-10}$ were concentrated and buffer exchanged into PBS using a 10 kDa Amicon. Non-fluorescent protein concentration was quantified by A280 using estimated extinction coefficients. GFP fusion protein concentration was quantified by A490 using ε = 117,000 M$^{-1}$cm$^{-1}$[42]. bdSUMO cleavage from GFP proteins was achieved by a 1–h 37 °C incubation with bdSENP1 in a molar ratio of 300:1 (GFP:bdSENP1) in cleavage buffer (500 mM NaCl, 50 mM Tris-HCl, 250 mM sucrose, 2 mM MgSO$_4$, 2 mM DTT, pH 8.0). bdSUMO, bdSENP1 and uncleaved GFP fusion proteins were separated by IMAC. Cleaved GFP proteins were buffer exchanged into buffer PB and concentrated using an Amicon 10 kDa MWCO ultrafiltration device. Cleaved ZF5.3 was buffer exchanged into buffer PB supplemented with 100 μM ZnCl$_2$. Purity of the proteins was determined by performing SDS-PAGE under non-reducing conditions and MALDI mass spectrometry. Proteins expressed in ClearColi had LPS levels of <1 EU/mL. All proteins were filtered using a 0.22-μm syringe filter then aliquoted and snap frozen in liquid nitrogen and stored at −80 °C.

**MALDI mass spectrometry.** MALDI-MS was performed using a MALDI-TOF/TOF ultrafleXtreme (Bruker-Daltonics) equipped with a 1 kHz laser or a MALDI-7090 (Shimadzu) equipped with a 2 kHz laser, both operated in linear positive-ion mode. A total of 5000 shots were summed using a 100-μm laser diameter and a user optimised laser intensity. Samples were prepared using the matrix 3-(4-hydroxy-3,5-dimethoxyphenyl)prop-2-enoic acid (sinapinic acid) and dissolved in a mixture of TA30 (30% acetonitrile and 70% water with trifluoroacetic acid (0.1% v/v)). Then, 1 μL of each protein was initially combined with 10 μL of matrix before spotting 2 μL of the sample onto a ground steel plate or AnchorChip target. External calibration was achieved with bovine serum albumin (BSA) using the average mass of the [M + H]$^+$ m/z ~66.5 kDa and [M + 2H]$^{2+}$ m/z ~33.3 kDa (Supplementary Table 2). Acquired spectra were exported to the open-source software mMass for processing (Supplementary Tables 2 and 3).

**Fluorescence microscopy.** To visualise actin staining, HEK293-LSA cells were seeded in 8-well microscopy chambers (Ibidi) at 30,000 cells/well. After an overnight incubation, cells were treated with 1 μM SNAP-cell 647-SiR (New England Biolabs) for 30 min, then washed three times with fresh media and incubated another 30 min to remove excess substrate. Cells were fixed using 4% paraformaldehyde for 10 min and washed thrice with PBS. Hoechst 33342 (Thermo Fisher Scientific) was diluted 1:2000 in FluoroBrite DMEM (Thermo Fisher Scientific) supplemented with 10% FBS and added to fixed cells. Imaging was performed using Olympus IX83 deconvolution wide field microscope with a 60x silicone objective. All images were processed using SlideBook 6.0 and ImageJ Fiji 2.0.0 software[45].

To visualise cellular distribution of EEP-GFP-HiBiT proteins, HEK293-LSA and HeLa-LSA cells were seeded 1 day prior at 10,000 cells/well into black 96-well clear bottom plates. EEP-GFP-HiBiT fusion proteins were pre-diluted to 1 μM in DMEM supplemented with 10% FBS and added to cells for 4 h. Cells were then washed with cell media three times and stained with Hoechst for 10 min before finally being imaged live in phenol-red free 10% FBS DMEM using a Leica TCS SP8 confocal microscope (GFP: 488 nm excitation, 500–570 nm emission, Hoechst: 810 nm (two photon) excitation, 420–470 nm emission).

**Calcein uptake.** Wild-type HEK293 and HEK293-LSA cells were seeded at 40,000 cells/well in a 96-well plate 1 day prior to the experiment. The cells were treated with calcein (100 μg/mL) for 0.5, 1, and 2 h. The cells were washed twice in cold growth media, then once with cold PBS, and detached with 50 μL TrypLE for 10 min. Then, 100 μL 1% BSA in PBS was added to each well and the entire contents were transferred to a 96-well, V-bottom plate. The cells were spun at 400 g for 5 min and the supernatant discarded. The cell pellet was resuspended in 100 μL 1% BSA in PBS and analysed by flow cytometry (Stratedigm). Data processing was performed on FlowJo 10.

**Cell viability.** HEK293-LSA cells were seeded 1 day prior at 10,000 cells/well into black 96-well clear bottom plates. Cells were treated with 1 μM EEP-GFP-HiBiT proteins and free HiBiT peptide for 4 h. Cells were washed with cell media three times and stained with Hoechst 33342 and PI. Leica SP8 Lightning confocal microscope was used to image cells (Hoechst: 405 nm excitation, 420–470 nm emission, PI: 488 nm excitation, 500–600 nm emission) (Supplementary Fig. 5).

To obtain larger microscopic fields of view, montage images of cells were taken using Olympus IX83 deconvolution wide field microscope with a 20x silicone objective. All images were processed using SlideBook 6.0 and ImageJ Fiji 2.0.0 software.

**Activity of EEP-GFP-HiBiT fusion proteins**. To determine the activity of the EEP-GFP-HiBiT fusion proteins, 50 nM of purified LgBiT protein was combined with 0.5 nM EEP-GFP-HiBiT fusion proteins in a black 96-well clear bottom microplate. NanoGlo Live Cell substrate was added to each well and luminescence was measured on the IVIS at 10 min post substrate addition. Luminescent signal from EEP-GFP-HiBiT proteins was normalised to the reference HiBiT peptide signal (Fig. 3b).

**Degradation of EEP-GFP-HiBiT protein**. HEK293 cells were seeded at 30,000 cells/well in a 96-well plate 1 day prior to the experiment. Cells were pre-chilled on ice before adding TAT-GFP-HiBiT protein (1 μM) After 1-h incubation on ice, unbound proteins were washed away and cells were incubated at 37 °C for 0.5 or 4 h to allow internalisation to take place. Cells were then lysed with RIPA buffer (supplemented with protease inhibitor). As positive controls, untreated cell lysate was spiked with TAT-GFP-HiBiT or free HiBiT peptide. Cell lysates were run on SDS-PAGE gels and transferred onto nitrocellulose membrane (0.45 μm pore size). The membrane was incubated with LgBiT protein overnight, and NanoGlo Luciferase Assay substrate was added. The membrane was imaged on ChemiDoc Imaging System (BioRad). Densitometric analysis of the membrane was performed to determine the extent of protein degradation after 4 h. Amount of protein in cell lysate after 0.5 h incubation of TAT-GFP-HiBiT protein serves as no degradation control.

**SLEEQ assay for determining endosomal escape efficiency**. HEK293 cells stably expressing LSA constructs were seeded 1 day prior at 10,000 cells per well into black 96-well clear bottom microplates. EEP-GFP-HiBiT proteins were added to cells at a final concentration of 1 μM and incubated for 4 h. For matched association experiments, the concentrations EEP-GFP-HiBiT proteins were adjusted accordingly (HiBiT: 2 nM, GFP: 1 μM, R9: 2 nM, TAT: 4 nM, ZF5.3: 2 nM, E5TAT: 2 nM, 5.3: 2 nM, pHD118: 500 nM, pHlip: 1 μM and HA2: 300 nM). The cells were washed three times with fresh media (DMEM, no phenol red, supplemented with 10% FBS) to remove excess unbound protein.

To measure cytosolic signal, 25 μL of diluted NanoGlo live cell substrate and 75 μL fresh media (with equal volume of DMSO used for digitonin permeabilisation) were added to cells. Luminescence measurements were made at 10 min after substrate addition on IVIS Lumina II. Exposure time was set to 10 s. Immediately after measuring cytosolic signal, 50 μL of cell supernatant was carefully transferred without disturbing the cell layer into another black 96-well clear bottom microplate and its luminescent signal measured.

To measure total cellular association, cells were treated with 0.01% w/v digitonin after the three washes and incubated for 1 h at 37 °C before substrate addition and luminescence measurement.

Both cytosolic and total cellular association signal were adjusted by subtracting the blank signal (untreated cells) and normalised to their respective EEP-GFP-HiBiT activities (Fig. 3b). In addition, the signal in cell supernatant was also subtracted from the cytosolic signal to account for any small amount of LgBiT/HiBiT complex released into the supernatant. The equations are shown below:

$$\text{Normalised cytosolic signal} = \frac{\text{Average radiance(live cells)} - \text{average radiance(blank)}}{\text{Activity of protein}}$$

(2)

$$\text{Normalised total cellular association} = \frac{\text{Average radiance(permeabilised cells)} - \text{average radiance(blank)}}{\text{Activity of protein}}$$

(3)

Endosomal escape efficiency was determined by the ratio of cytosolic signal to total cellular association. The equation is shown below:

$$\text{Endosomal escape efficiency(\%)} = \frac{\text{Normalised cytosolic signal}}{\text{Normalised total cellular association}} \times 100\%$$

(4)

Endosomal escape efficiency was calculated for each independent experiment and averaged to provide mean ± SEM (Figs 4c, d and 6c, d).

The limit of quantification was defined as three times the standard deviation of the background signal (LSA expressing cells in the presence of substrate, without HiBiT)

**Endocytosis inhibition studies**. HEK293-LSA cells were seeded 1 day prior at 10,000 cells per well into black 96-well clear bottom plates. Cells were incubated at 4 °C or treated with endocytosis inhibitors (50 μM EIPA, 10 mM NaN₃ + 10 mM 2-deoxy-D-glucose, 500 nM cytochalasin D, 50 nM wortmannin) for 20 min before adding 1 μM GFP-HiBiT protein. After 4-h incubation, cells were washed three times and replaced with fresh media. NanoGlo Live Cell Substrate was added and

50 μL of cell media was removed and placed into a fresh 96-well plate for luminescence measurement.

**Statistical analysis**. Luminescence data were processed on GraphPad Prism 7 software and presented as mean ± standard error of mean. Statistical analyses were performed using Kruskal–Wallis with uncorrected Dunn's test, with GFP being the control group. Sample sizes ($n$) are provided in the respective figure legends. Asterisks represent statistical significance (ns denotes $p > 0.05$, * denotes $p \leq 0.05$, ** denotes $p \leq 0.01$, *** denotes $p \leq 0.001$ and **** denotes $p \leq 0.0001$).

**Reporting summary**. Further information on research design is available in the Nature Research Reporting Summary linked to this article.

## Data availability
Source data for all the figures are provided with the paper. The raw data (microscopy images, luminescent images and mass spectrometry files) that support the findings of this study are available from the corresponding author upon reasonable request. Source Data are provided with this paper.

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

## Acknowledgements

MALDI was performed at the Melbourne Centre for Nanofabrication (MCN) in the Victorian Node of the Australian National Fabrication Facility (ANFF). We thank Dr David Rudd for providing his MALDI expertise. S. L. Y. T. was supported by Monash Graduate Scholarship (MGS). J. J. R. was supported by an Australian Government Research Training Program (RTP) Scholarship.

## Author contributions

C. W. P. and A. P. R. J. developed the concept and supervised the study. S. L. Y. T. performed fluorescence imaging for visualising actin filaments, designed and performed all of the luminescence experiments, analysed the data and designed the figures. J. J. R. assisted with cloning, virus production, prepared and characterised the fusion proteins and performed the imaging for visualisation protein distribution. H. A.-W. provided assistance in virus production and developing the concept. D. Y. provided assistance in designing molecular constructs of the fusion proteins. S. L. Y. T. wrote the manuscript, J. J. R., D. Y., C. W. P. and A. P. R. J. edited the manuscript.

## Competing interests

The authors declare no competing interests.
