## [Peer Review File · Nature Communications]

Reviewers' Comments:

Reviewer #1:

Remarks to the Author:

The work presented by Pouton and co-workers addresses an important challenge: how to quantify precisely how much proteins enter cells, in particular through the activity of endosomal escape agent. This is important because 1) endosomal entrapment is a bottleneck on the delivery field, 2) there is controversy in field because numerous potential pitfalls and qualitative assays, 3) mechanistic insights are key to the design of improved systems.

Pouton and co-workers pursue a split luciferase approach: the cell entry of one fragment HiBiT complements a partner LgBiT fragment already in the cells.

The work is well executed in a number of experiments and well written.

Some of the conclusions potentially controversial but useful.

Nonetheless, there are several issues with the manuscript in its current form:

1) The use of LgBiT/HiBiT has been recently reported to study the cell penetration of EN2 (Joliot, A. bidirectional transfer of engrailed homeoprotein across the plasma membrane requires PIP2, bioRxiv, 2020).

The timing is unfortunate but it nonetheless puts in question the novelty of the approach.

2) The authors do not formally demonstrate that endosomal escape is involved (rather than plasma membrane translocation or other unknown mechanisms). I do agree that it is likely involved, but plasma membrane translocation is not directly ruled out (to be clear, there is probe inside endosomes, but it is not clear that the little bit that gets in the cytosol is coming from endosomes).

3) The authors use 1 μ M proteins. On one hand, it makes sense as they test conditions that are maybe relevant to therapeutic applications. On the other hand, it is well established that membrane active peptides often need to reach a threshold concentration to induce permeabilization (in endosomes or elsewhere). In this context, seeing low improvement on escape can be expected and this could be presented in a more nuanced way than "they help with endocytosis but otherwise don't work".

4) E5-TAT has been shown to induce endosomal escape while itself remaining trapped in the membrane of disrupted endosomes (same group as ref 31, BBA paper). The authors hint at such potential issues in their discussion but it is not clear. In any case, this would be an example of something that does promote endosomal escape, simply not very well for itself or for what is attached to it. This additional complexity should be discussed.

5) Detecting low levels of escape come with the challenge of "noise" signal from artifact. The authors try to address this. However, I am concerned about cell death: Figure S4 is not quantitative and cell death often occurs in regular cell culture (maybe 5% or less??). My point here is that just a few dead cells could contribute a lot of signal and confound conclusions.

6) This is a bulk assay. It is possible that a few cells contribute to the majority of the successful HiBiT/LgBiT luminescence (maybe there are a few sick but not dead cells, or cells that have somewhat damaged membranes because). It is a very different situation than if all cells have just a little bit of signal. This should be addressed because it impacts the conclusions.

7) proteins expressed with cationic CPPs tend to purify with LPS from E.coli or other anionic molecules (work by Futaki). the "copious" washing used need to be shown as effective in removing such potential disruptive contaminants. Assessing purity of the proteins by SDS-PAGE/MS of protein is not enough to guarantee no interferences.

8) I am unclear on how "cellular association" is determined. From the methods, it seems that plasma membrane and endosomal membranes are permeabilized and that substrate is then added. 1) how do you know that all endosomes are permeabilized, 2) how do you know that all HiBit find and react with LgBit (do you add excess LgBit)?....Finally, how do you distinguish plasma membrane bound and endosomal fractions? Is the luminescence of what is extracellular tested for every sample (sorry, I had trouble finding the details)?

This is pretty important as the conclusions rely heavily on this measurement.

The experiment comparing similar levels of EEP-probe associated with cells is also problematic if a significant portion of the signal is extracellular (I understand that the washes are stringent, but as reported and shown here for E5-TAT, it may not be enough to remove plasma membrane bound probe, figS6?).

9) there is another level of complexity worth thinking about: unfolded peptides can get rapidly degraded while transiting in the endocytic pathway. The HiBiT tag on the GFP payload could be a target of degradation. If so, this could lead to underestimations of how much is released or entrapped (maybe even more for the portion that remains trapped and that is subjected to degradation for longer; although cytosolic degradation will still occur).

Importantly, GFP with a cleaved HiBit could still get into the cytosol (because it is folded, it will likely have a longer half life). In a worst case scenario, the EEP also change where the payloads go (different endosomes?), and because of that, degradation and results are different.

10) the PHlip sequence in Supp info is the TAT sequence

11) discussion "Fig.s 5 and S5". typo and i think you mean S6.

Reviewer #2:

Remarks to the Author:

The authors present an exceptionally well-written manuscript describing solid, well-done research engineering a solution to a vexing problem in the cell-penetrating peptide field, the likely impact of which is significant enough to warrant publication in Nature Communications. The field has long been needing a quantitative assay for endosomal escape. The authors' fusion of LgBiT to actin is highly innovative, overcoming what was to be my objection before I could make it that secretion and subsequent peptide binding followed by re-endocytosing of the fluor could be an artifact.

It is likely that the field will be surprised at the finding that endosomal escape is so poor that CPPs are generally no better than constitutive transport when it comes to delivering to the cytoplasm. I think it a strength of the paper that the authors, for the first time, quantitatively demonstrate exactly how poor endosomal escape is.

Use of "active transport" in several places should be clarified as the authors mean "active endosomal escape mechanism;" cellular entry may well proceed via an active transport process such as receptor-mediated endocytosis or macropinocytosis.

The figures are well done and appropriately presented. Methods are sufficiently detailed. Supplemental material is both appropriately supplemental and useful for the reader who wishes to attempt to replicate.

I have never before recommended a paper to accept as is on a first submission, but that is nevertheless my recommendation. I have only very minor quibbles with small points in writing, namely:

lines 48-49: I don't think all CPPs are EEPs (and they certainly, as the authors quantitatively measure, are much better at penetrating the cell than escaping the endosome!). There are many other papers that draw a distinction between CPP moieties and EEE (endosomal escape enhancement) moieties such as dFTAT, GALA or Aurein. I encourage the authors to reword this.

lines 135 and 342: it's been a long time since I had a grammar class, so I'm not sure if these sentences constitute run-ons, but in any event, they would better be broken into two sentences with a ". However,..." rather than "... , however ...".

line 167: "exhibits" should be "exhibit"

Reviewer #3:

Remarks to the Author:

This manuscript describes a NanoBit-based luminescent assay to assess the relative potency of a variety of previously reported cell penetrating peptides. The assay is very sensitive, which is great, can in theory be performed in high-throughput, and overall the experiments are well controlled. The paper is exceptionally well written. There are a few major issues that affect the conclusions and should be addressed before publication.

1. The scholarship is low. The field has progressed significantly since many of the cited foundational articles were published. Most notably, the recent work of Kritzer is not cited or discussed. The Kritzer assay is also very sensitive and avoids some of the limitations of SLEEQ.
2. The premise is not accurate. There is a published method that quantifies endosomal release directly. What's worse, it is cited but not mentioned. This method, which is based on single molecule spectroscopy, is more direct than the method described here and equally sensitive. It is also not subjective, and provides some evidence that the material being observed is not degraded (a huge smoking gun for SLEEQ and most other methods, given the plethora of proteases present within living cells).
3. Controls are missing. The signal upon which this assay relies will appear even if the cargo – the material attached to HiBiT – is partially or completely degraded. Since the extent of degradation was not evaluated biochemically – which is laborious, but can be done – the comparisons made in the paper are not valid and no conclusions can be drawn from them. The authors could simply be comparing how much HiBiT is released from their materials in endolysosomes during a 4 h incubation. This is a long time!
4. Controls are missing – the conclusions drawn also assume that the affinity of LgBiT and HiBiT could be affected by either the presence of GFP or the CPP. This could also affect the conclusions drawn, but was not addressed. The activity data shown in Figure 3 is not sufficient – one needs the K_d for the HiBiT/LgBiT interaction for each construct.
5. Time is an important variable. Most studies evaluate penetration as a function of time and concentration. This is especially important when proteolytic sensitivity is not assessed.
6. The ultimate conclusion "These results fundamentally change our understanding of the role of EEPs and how they promote cytosolic delivery." Is not fully supported by the data.

Response to reviews comments

Reviewer 1

1) The use of LgBiT/HiBit has been recently reported to study the cell penetration of EN2 (Joliot, A. bidirectional transfer of engrailed homeoprotein across the plasma membrane requires PIP2, bioRxiv, 2020).

The timing is unfortunate but it nonetheless puts in question the novelty of the approach.

We thank the reviewer for pointing out this pre-print article, and we have included it as a reference in the revised version of the manuscript. While the bioRxiv article uses the LgBiT/HiBiT assay, the focus of the paper is specifically on the translocation of EN2 across the plasma membrane, and the role of PIP2 in this process. The paper is not focused on investigating endosomal escape and it does not measure endosomal escape efficiency.

The significance of our paper is two fold. First, we have developed the SLEEQ assay, which for the first time allows us to directly quantify both the amount of exogenous protein delivered to the cytosol, and the endosomal escape efficiency of these proteins. Existing assays (including the bioRxiv article) do not measure the efficiency of escape, which is critical to understanding how these materials work.

Second, we have screened 8 commonly used endosomal escape peptides and demonstrated that cationic peptides increase delivery to the cytosol simply in increasing association to the cell. This is a major new finding and has significant implications to the design of endosomal escape agents.

2) The authors do not formally demonstrate that endosomal escape is involved (rather than plasma membrane translocation or other unknown mechanisms). I do agree that it is likely involved, but plasma membrane translocation is not directly ruled out (to be clear, there is probe inside endosomes, but it is not clear that the little bit that gets in the cytosol is coming from endosomes).

The reviewer raises a valid point that the cytosolic delivery we observe could be due to direct plasma membrane translocation. While it is unlikely that a charged, >30kDa protein is able to translocate across the membrane, we cannot rule out this possibility, so we have revised the manuscript as follows to clarify this point:

“However, it cannot be ruled out that the small amount of protein that reaches the cytosol could be the result of translocation across the plasma membrane.” (page 21)

3) The authors use 1 uM proteins. On one hand, it makes sense as they test conditions that are maybe relevant to therapeutic applications. On the other hand, it is well established that membrane active peptides often need to reach a threshold concentration to induce permeabilization (in endosomes or elsewhere). In this context, seeing low improvement on escape can be expected and this could be presented in a more nuanced way than "they help with endocytosis but otherwise don't work".

The reviewer is correct that a number of EEPs have been reported to require high concentrations of peptide to observe significant levels of escape. To address this point, we have performed additional experiments with concentrations of protein up to 10 μ M, and observed a similar trend to the result at 1 μ M (Supplementary Fig. 9). We have revised the manuscript as follows to reflect these additional experiments:

“It has been suggested that higher concentrations of EEPs may be required to induce endosomal escape. No significant increase in endosomal escape efficiency was observed when HEK293 cells were treated with 10 μ M of EEP-GFP-HiBiT (Supplementary Fig. 9). (p 18)”

While it would be possible to test the proteins at even higher concentrations, as the reviewer noted, these high concentrations are not therapeutically relevant. A significant advantage of our SLEEQ assay is its ability to sense pM concentrations of protein, and measure endosomal escape at therapeutically relevant concentrations.

4) E5-TAT has been shown to induce endosomal escape while itself remaining trapped in the membrane of disrupted endosomes (same group as ref 31, BBA paper). The authors hint at such potential issues in their discussion but it is not clear. In any case, this would be an example of something that does promote endosomal escape, simply not very well for itself or for what is attached to it. This additional complexity should be discussed.

We agree that it is possible that the EEPs may disrupt the membrane but not escape from the endosome themselves. We have revised this section as follows to make this point clearer:

“These experiments do not exclude the possibility that EEPs delivered separately from the cargo protein (i.e not as a fusion protein but as a separate endosomal escape agent) could promote endosomal escape. However, delivery of an EEP separately from the intended therapeutic would induce non-specific endosomal escape in a range of cells, which could lead to significant side effects. Therefore it is highly desirable for EEPs to be functional as a fusion protein.” (page 21)

5) Detecting low levels of escape come with the challenge of "noise" signal from artifact. The authors try to address this. However, I am concerned about cell death: Figure S4 is not quantitative and cell death often occurs in regular cell culture (maybe 5% or less??). My point here is that just a few dead cells could contribute a lot of signal and confound conclusions.

The reviewer is correct that cell death could create a ‘noise’ artifact in our measurements, however the way we have developed the assay should minimise this possibility. After incubating the cells with protein, before taking the measurement, the supernatant is removed. This will likely dislodge any sick or dying cells, eliminating them from our measurement. We have included additional images of the cells after the assay, to demonstrate the viability of the remaining cells is significantly higher than 99% (Supplementary Fig. 5).

Additionally, we measure the luminescence of the cell supernatant, which would contain LgBiT excreted by any dead or dying cells. For all the samples, the signal from the supernatant was less than 0.2% (Figure 2b), which indicates minimal excretion of LgBiT into the cytosol, and minimal cell death.

We have revised the manuscript as follows to make this point clearer. “Signal in the cell supernatant was also sampled to correct for any cell death that could contribute to false positive signals.” (page 12)

Furthermore, even if there is some signal due to dead or dying cells, this would cause us to over-estimate the endosomal escape. Given that the levels of endosomal escape are very low, measuring even lower levels of endosomal escape would not affect the conclusions we have made in this paper.

6) *This is a bulk assay. It is possible that a few cells contribute to the majority of the successful HiBit/LgBit luminescence (maybe there are a few sick but not dead cells, or cells that have somewhat damaged membranes because). It is a very different situation than if all cells have just a little bit of signal. This should be addressed because it impacts the conclusions.*

As per our response to point 5 above, we have taken this into account, and revised the manuscript to make this point clearer.

7) *proteins expressed with cationic CPPs tend to purify with LPS from E.coli or other anionic molecules (work by Futaki). the "copious" washing used need to be shown as effective in removing such potential disruptive contaminants. Assessing purity of the proteins by SDS-PAGE/MS of protein is not enough to guarantee no interferences.*

The reviewer is correct to say that it is possible that there could be some LPS in our purified protein, however it is unlikely that LPS would interfere with the assay. To demonstrate this, we added LPS to the SLEEQ assay and we saw no major effect on the endosomal escape results.

8) *I am unclear on how "cellular association" is determined. From the methods, it seems that plasma membrane and endosomal membranes are permeabilized and that substrate is then added. 1) how do you know that all endosomes are permeabilized, 2) how do you know that all HiBit find and react with LgBit (do you add excess LgBit)?....Finally, how do you distinguish plasma membrane bound and endosomal fractions? Is the luminescence of what is extracellular tested for every sample (sorry, I had trouble finding the details)? This is pretty important as the conclusions rely heavily on this measurement.*

To demonstrate that digitonin permeabilises all the endosomes, we have included additional data (Supplementary Fig. 3) which shows cells incubated with calcein (a membrane impermeable dye that is endocytosed into endosomes) before and after digitonin treatment. From this figure, it is clear that all calcein signal has been lost, indicating that the endosomes have been disrupted.

In Supplementary Fig. 2, we demonstrate the cells have an excess of LgBiT, therefore it was not necessary to add additional LgBiT to the assay. Furthermore, if not all the HiBiT can find LgBiT, this would cause us to under-estimate the total association, leading to an over-estimation of endosomal escape efficiency. As pointed out in response to point 5, given that the levels of endosomal escape are very low, measuring even lower levels of endosomal escape efficiency would not affect the conclusions we have made in this paper.

The experiment comparing similar levels of EEP-probe associated with cells is also problematic if a significant portion of the signal is extracellular (I understand that the washes are stringent, but as reported and shown here for E5-TAT, it may not be enough to remove plasma membrane bound probe, figS6?).

We disagree with the reviewer on this point.

Membrane bound protein will not interfere with out cytosolic measurement of protein, as there is no LgBiT present on the surface of cells to give a luminescent signal. Membrane bound protein will be measured in our digitonin treated samples, and we want it to be, as this is part of the total amount of protein associated with the cells. The digitonin treated sample measures the total association of protein (regardless of if it is in the cytosol, in endosomes or on the plasma membrane).

9) there is another level of complexity worth thinking about: unfolded peptides can get rapidly degraded while transiting in the endocytic pathway. The HiBiT tag on the GFP payload could be a target of degradation. If so, this could lead to underestimations of how much is released or entrapped (maybe even more for the portion that remains trapped and that is subjected to degradation for longer; although cytosolic degradation will still occur). Importantly, GFP with a cleaved HiBit could still get into the cytosol (because it is folded, it will likely have a longer half life). In a worst case scenario, the EEP also change where the payloads go (different endosomes?), and because of that, degradation and results are different.

One of the strengths of our SLEEQ assay is that it only measures signal from active protein. If the protein is degraded then no-signal will be detected. For the therapeutic application of EEPs, we are only interested if in-tact protein is delivered to the cytosol, not degraded protein. This is an important distinction from conventional fluorescently tagged materials, where the fluorescent dye can be cleaved from the protein, and the signal measured could be due to free dye.

To demonstrate that HiBiT isn't cleaved from the GFP we performed an additional experiment, where cells were incubated with HiBiT-GFP at 4°C for an hour, followed by a 4 hour incubation at 37°C. The cells were then lysed in the presence of a protease inhibitor and the cell lysate run on an SDS-PAGE gel. HiBiT was detected by imaging the gel in the presence of LgBiT (Supplementary Fig 10). As shown by the gel, a single band at the same molecular weight of HiBiT-GFP was observed for both the 0.5 hr and 4hr incubation periods.

No low molecular band, or smearing of the protein band was observed. This demonstrates that HiBiT is not cleaved from the HiBiT-GFP, and we are indeed measuring the endosomal escape of the whole protein. We have revised the manuscript as follows to reflect this additional experiment.

“Western blot analysis of cell lysate (Supplementary Fig. 10) shows that the fusion protein largely remains intact. More importantly, no low molecular weight band corresponding to free HiBiT peptide was observed, indicating that the luminescent signal measured in the SLEEQ assay comes from intact fusion protein, and not degradation products of the protein.” (page 20)

10) the PHlip sequence in Supp info is the TAT sequence

Thank you for pointing this out, we have corrected this typo.

11) discussion "Fig.s 5 and S5". typo and i think you mean S6.

The reviewer is correct and we have corrected this typo.

Reviewer 2

The authors present an exceptionally well-written manuscript describing solid, well-done research engineering a solution to a vexing problem in the cell-penetrating peptide field, the likely impact of which is significant enough to warrant publication in Nature Communications. The field has long been needing a quantitative assay for endosomal escape. The authors' fusion of LgBiT to actin is highly innovative, overcoming what was to be my objection before I could make it that secretion and subsequent peptide binding followed by re-endocytosing of the fluor could be an artifact.

It is likely that the field will be surprised at the finding that endosomal escape is so poor that CPPs are generally no better than constitutive transport when it comes to delivering to the cytoplasm. I think it a strength of the paper that the authors, for the first time, quantitatively demonstrate exactly how poor endosomal escape is.

Use of "active transport" in several places should be clarified as the authors mean "active endosomal escape mechanism;" cellular entry may well proceed via an active transport process such as receptor-mediated endocytosis or macropinocytosis.

We thank the reviewer for their positive comments and have revised the manuscript to clarify the use of the term 'active transport'. We have revised the manuscript as follows:

“This suggests that the positively charged EEPs increase cytosolic accumulation mostly through non-specific association with the cells, rather than enhancing the efficiency of endosomal escape.” (page 5)

“Rather than improving endosomal escape efficiency, it appears that when positively charged EEPs are fused to a protein, they increase cytosolic delivery simply by increasing cellular accumulation.” (page 21)

“Our findings demonstrate that while positive EEPs improved cytosolic accumulation, it was not via enhancement of endosomal escape.” (page 22)

The figures are well done and appropriately presented. Methods are sufficiently detailed. Supplemental material is both appropriately supplemental and useful for the reader who wishes to attempt to replicate.

I have never before recommended a paper to accept as is on a first submission, but that is nevertheless my recommendation. I have only very minor quibbles with small points in writing, namely:

lines 48-49: I don't think all CPPs are EEPs (and they certainly, as the authors quantitatively measure, are much better at penetrating the cell than escaping the endosome!). There are many other papers that draw a distinction between CPP moieties and EEE (endosomal escape enhancement) moieties such as dfTAT, GALA or Aurein. I encourage the authors to reword this.

We thank the reviewer for this suggestion and have reworded this section as follows:

“During the last 30 years, endosomal escape peptides (EEPs), which are a subclass of cell-penetrating peptides (CPPs), have emerged as promising delivery agents for enhancing endosomal escape.” (page 3)

lines 135 and 342: it's been a long time since I had a grammar class, so I'm not sure if these sentences constitute run-ons, but in any event, they would better be broken into two sentences with a ". However,..." rather than "... , however ...".

Again, we thank the reviewer for this comment and have revised the manuscript as follows:

“Split GFP systems have previously been employed to detect cytosolic delivery of EEPs^{16,17,19}. However, the sensitivity of fluorescence techniques is typically limited to micromolar concentrations.” (page 7)

“The process of endosomal escape is postulated to be highly inefficient. However until now, our estimates of endosomal escape efficiency have been qualitative.” (page 18)

line 167: "exhibits" should be "exhibit"

We have corrected this typo.

“These results demonstrate that both HiBiT peptide and LgBiT protein exhibit minimal background signal on their own.” (page 9)

Reviewer 3

This manuscript describes a NanoBit-based luminescent assay to assess the relative potency of a variety of previously reported cell penetrating peptides. The assay is very sensitive, which is great, can in theory be performed in high-throughput, and overall the experiments are well controlled. The paper is exceptionally well written. There are a few major issues that affect the conclusions and should be addressed before publication.

1. The scholarship is low. The field has progressed significantly since many of the cited foundational articles were published. Most notably, the recent work of Kritzer is not cited or discussed. The Kritzer assay is also very sensitive and avoids some of the limitations of SLEEQ.

We and the other reviewers (especially reviewer 2) disagree that the scholarship is low.

In the original manuscript, we only included “switch on” endosomal escape sensors. We have revised the manuscript to expand this discussion to include more general fluorescence methods including the work of Kritzer. However, we note that the Kritzer assay is fluorescence based, and therefore is significantly less sensitive than the SLEEQ assay. Additionally, the Kritzer assay does not measure the endosomal escape efficiency of molecule, but indirectly measures the amount of material delivered to the cytosol through a blocking assay. This is an important distinction to the SLEEQ assay. For example, the Kritzer assay cannot determine endosomal escape efficiency, and therefore would not have discovered that increased cytosolic accumulation of positively charged EEPs is due to membrane association. We have revised the manuscript as follows:

“To overcome this, techniques such as the chloroalkane penetration assay and fluorescence correlation spectroscopy (FCS) provide a way of measuring the amount of material that has reached the cytosol, but do not measure the efficiency of endosomal escape.” (p 19)

It is not clear what advantages the Kritzer assay has over the SLEEQ assay, but we are happy to respond to these if the reviewer outlines what they are.

2. The premise is not accurate. There is a published method that quantifies endosomal release directly. What’s worse, it is cited but not mentioned. This method, which is based on single molecule spectroscopy, is more direct than the method described here and equally sensitive. It is also not subjective, and provides some evidence that the material being observed is not degraded (a huge smoking gun for SLEEQ and most other methods, given the plethora of proteases present within living cells).

We assume the reviewer is referring to the use of fluorescence correlation spectroscopy

FCS differs from SLEEQ in a number of key ways. First, SLEEQ is significantly more sensitive. The sensitivity of FCS is ~1nM, whereas we have shown SLEEQ to be sensitive to 5 pM (Figure 2a). Second, FCS is inherently low throughput. Third, FCS is subjective, as it is challenging to demonstrate that the area where the FCS measurement is taken is truly cytosolic. The cytoplasm is filled with thousands of highly dynamic vesicles and ensuring the measurement is performed in a way that completely excludes any of these compartments is

challenging. Finally, FCS does not all an easy way to quantify total association, and therefore cannot determine endosomal escape efficiency, as demonstrated by the SLEEQ assay. We have included reference to FCS techniques in the revised manuscript, as outlined above.

With regards to the presence of proteases and the potential degradation of protein, SLEEQ offers a significant advantage to techniques that seek to detect fluorescently labelled materials. As outlined in response to reviewer 1, for therapeutic applications we are only interested in the delivery of in-tact protein to the cytosol, not degraded material. There is the possibility that proteins tagged with fluorescent dyes are degraded and signal detected in the cytosol is dye cleaved from the protein. SLEEQ overcomes this limitation and requires the protein to be active and intact for the complementation with LgBiT and luminescence to occur. We have demonstrated that there is no detectible levels of free HiBiT in lysed cells (Supplementary Fig. 10) and revised the manuscript as follows:

“Western blot analysis of cell lysate (Supplementary Fig. 10) shows that the fusion protein largely remains intact. More importantly, no low molecular weight band corresponding to free HiBiT peptide was observed, indicating that the luminescent signal measured in the SLEEQ assay comes from intact fusion protein, and not degradation products of the protein.” (p 20)

3. Controls are missing. The signal upon which this assay relies will appear even if the cargo – the material attached to HiBiT – is partially or completely degraded. Since the extent of degradation was not evaluated biochemically – which is laborious, but can be done – the comparisons made in the paper are not valid and no conclusions can be drawn from them. The authors could simply be comparing how much HiBiT is released from their materials in endolysosomes during a 4 h incubation. This is a long time!

To address this point, we lysed cells after 4 hour incubation with the HiBiT-GFP. As shown in Supplementary Fig. 10, there are no detectable levels of free HiBiT, which demonstrates the signal we are measuring from our assay comes from complete protein. It is possible that some HiBiT-GFP is degraded in endosomes. As outlined in our response to point 2 above, the advantage of the SLEEQ assay is degraded protein will not be detected by the assay.

4. Controls are missing – the conclusions drawn also assume that the affinity of LgBiT and HiBiT could be affected by either the presence of GFP or the CPP. This could also affect the conclusions drawn, but was not addressed. The activity data shown in Figure 3 is not sufficient – one needs the Kd for the HiBiT/LgBiT interaction for each construct.

We disagree with the reviewer that Kd measurements will provide useful additional information about the assay.

All our luminescence measurements are dependent on the activity of the HiBiT/LgBiT construct. Luminescence is the product not only of the Kd of the HiBiT/LgBiT interaction, but also the light generating efficiency of the completed enzyme. The measurements in Figure 3 take both these factors into account. A Kd measurement does not take into account if the EEP or GFP affect the activity of the luciferase.

5. Time is an important variable. Most studies evaluate penetration as a function of time and concentration. This is especially important when proteolytic sensitivity is not assessed.

A time course study could be the focus of future studies but would not add to the conclusion that we have made to this paper, which is EEPs increase cytosolic accumulation due to increased cellular association, not improved endosomal escape efficiency. In the updated manuscript we have demonstrated that only intact protein is measured in the assay.

6. The ultimate conclusion “These results fundamentally change our understanding of the role of EEPs and how they promote cytosolic delivery.” Is not fully supported by the data.

We disagree with this assertion. Our results show that EEP-GFP proteins increased both cytosolic delivery and total cellular association at two different concentrations (1 and 10 μ M) (Figure 4 and Supplementary Fig. 9), which suggested that the increase in cytosolic protein was a result of increased cellular accumulation. To further demonstrate this theory, we tested the proteins at different concentrations such that the total cellular association was similar among all the proteins (Figure 6). We observed no significant increase in cytosolic delivery compared to GFP alone, indicating that the EEPs do not enhance endosomal escape.

Much of the current dogma around EEPs is that these peptides play a significant role in improving endosomal escape efficiency. However, while we found that some EEP fusion proteins exhibited higher levels of cytosolic delivery, the increased cytosolic delivery was the result of increased association with cells.

Reviewers' Comments:

Reviewer #1:

Remarks to the Author:

The authors have made a significant efforts to address my comments and I thank them for that. I still have a couple of disagreements: comment 7) I would expect LPS to be inhibitory to endosomal escape so I'm not clear on the rationale of adding it to prove that it does not interfere (increasing inhibition of a non-escape signal?), comment 9) I had envisioned HiBit being degraded from GFP without GFP being degraded (because GFP is fairly stable): this would lead to an underestimation of the protein delivered (it may not be intact but it is the "payload" part that people would care about delivering, having HiBiT attached to it doesn't matter once it is inside cells)...the authors checked degradation but I'm unclear on whether they would detect GFP vs HiBiT-GFP (given that HiBiT is small). Also, if HiBit is cleaved from GFP, I would expect it to be rapidly and completely degraded and hence not detected (so not seeing it doesn't mean that no degradation took place, and GFP would still be around). Finally, there is still the possibility of confusion for a peptide like E5-TAT. Based on our experience, it does promote endosomal escape...in trans, not in cis (this peptide does have problems with hydrophobicity, aggregation, and cell surface sticking). I understand the point of view of the authors ("fusion proteins is what we care about"), but I'm just pointed out the fact that there is more nuance than "E5-TATs don't work" (which is what I think people will remember from this paper, and which is somewhat misleading).

The important thing here is that readers are made aware of these issues and that they have the info they need to draw their own conclusions. It is now the case.

Jean-Philippe Pellois

Reviewer #3:

Remarks to the Author:

The authors have gone to great lengths to respond to the comments of the previous reviewers, in some cases successfully. Here are the concerns that unfortunately still remain:

1. (minor) I do not believe that the authors have successfully ruled out that some fraction of the signal they observe derives from direct translocation across the plasma membrane – the response to reviewer 1 is not ideal. The decreases observed at 4 °C could be due simply to changes in membrane fluidity, as the authors note. There are other ways to inhibit endocytosis, including with a cell permeable dynamin inhibitor.
2. The authors go to great length to discredit FCS as a valid technique, but do not include any experiments or literature citations to support their assertion "it is challenging to demonstrate that the area where the FCS measurement is taken is truly cytosolic". FCS is low throughput, but it is the gold standard, and should be used by the authors to validate their assay (see below). Doing so would greatly improve this manuscript.
3. The evidence provided to suggest that the delivered material is not degraded is insufficient. The control provided in the rebuttal (gel of total cell lysate, not isolated cytosol) is not enough given the low efficiency of escape of the GFP conjugates. The conclusion drawn from the absence of HiBiT peptide is even worse (as the absence of a signal proves little – a classic "negative result". The stability of the HiBiT peptide under the experimental conditions is not included as a control.
4. No secondary validation. Assays need validation – using a second technique. How can a paper for a "new assay" be published without some form of secondary validation?
5. The assay demands fusion of the protein being delivered to GFP. It is well known that cargo identity (in this case GFP) affects delivery. Do the authors propose to fuse another protein to GFP

to execute the assay on a meaningful therapeutic target? What is GFP inhibits the process or processes that are required for endosomal release? See below for a better use of luminescence for monitoring localization.

6. Prior art. The timing is unfortunate, but this paper has now been subverted at least in part by the Merck group (<https://pubs.acs.org/doi/10.1021/acscchembio.0c00804>). This assay uses an azide tag (N₃ – far less perturbing than GFP) and is just as sensitive.

Reviewer #1:

The authors have made a significant efforts to address my comments and I thank them for that. I still have a couple of disagreements:

Comment 7: *I would expect LPS to be inhibitory to endosomal escape so I'm not clear on the rationale of adding it to prove that it does not interfere (increasing inhibition of a non-escape signal?)*

To address the reviewers concern about the potential of LPS to interfere with endosomal escape we have repeated the SLEEQ experiments with protein made in ClearColi. ClearColi is a strain of E. Coli does not contain LPS. These experiments showed no difference in the endosomal escape of protein expressed in ClearColi or E. Coli.

We have added the following sentence to the manuscript along with SI Figure 12.

"It has also been proposed that the presence of lipopolysaccharides (LPS) from the bacterial used to express the EEP-GFP-HiBiT could interfere with endosomal escape. To control for this, we performed the SLEEQ assay using protein synthesised in ClearColi, a strain of E. Coli does not contain LPS (Supplementary Fig. 12). We observed no difference in the endosomal escape efficiency of GFP-HiBiT and TAT-GFP-HiBiT synthesised in ClearColi or E. Coli."

Comment 9: *I had envisioned HiBit being degraded from GFP without GFP being degraded (because GFP is fairly stable): this would lead to an underestimation of the protein delivered (it may not be intact but it is the "payload" part that people would care about delivering, having HiBiT attached to it doesn't matter once it is inside cells)...the authors checked degradation but I'm unclear on whether they would detect GFP vs HiBiT-GFP (given that HiBiT is small). Also, if HiBit is cleaved from GFP, I would expect it to be rapidly and completely degraded and hence not detected (so not seeing it doesn't mean that no degradation took place, and GFP would still be around).*

To address the reviewers concern about HiBit being degraded while GFP remains intact, we performed additional experiments on the cell lysate to look for loss of HiBiT signal over time. We saw no significant change in the HiBiT over 4 hours (SI Figure 8b), showing that there is not significant degradation of HiBiT.

From the degradation experiments we can conclude that:

- 1) There is no detectable degradation of GFP-HiBiT over the course of the experiment, therefore we are not underestimating the amount of protein delivered
- 2) Free HiBiT is not released from the GFP-HiBiT, therefore any signal detected in the cells is due to the full protein and not degradation products.

We have revised the manuscript as follows to reflect this additional experiment.

"and no degradation of the GFP-HiBiT fusion was detected over the period of the experiment (Supplementary Fig. 8b)."

Finally, *there is still the possibility of confusion for a peptide like E5-TAT. Based on our experience, it does promote endosomal escape...in trans, not in cis (this peptide does have problems with hydrophobicity, aggregation, and cell surface sticking). I understand the point of view of the authors ("fusion proteins is what we care about"), but I'm just pointed out the fact that there is more nuance than "EEPs don't work" (which is what I think people will remember from this paper, and which is somewhat misleading).*

It was not our intention to say that peptides like E5-TAT do not have any endosomal escape properties by themselves. The intention of the paper is to highlight the widespread misconception that fusing an EEP to a protein will result in improved efficiency of delivery to the cytosol. We have further revised this section as follows to address this point.

“These experiments focus on EEPs fused to a cargo and do not exclude the possibility that EEPs delivered separately from the cargo protein (i.e not as a fusion protein but as a separate endosomal escape agent) or when fused to a short peptide could promote endosomal escape.”

Reviewer #3:

1) The authors have not successfully ruled out that some fraction of the signal they observe derives from direct translocation across the plasma membrane. There are other ways to inhibit endocytosis, including with a cell permeable dynamin inhibitor.

Dynamin inhibition does not stop macropinocytosis (the most likely mechanism of non-specific protein uptake) or the CLIC/GEEC pathway, so use of these inhibitors would not rule out direct translocation across the membrane. In the previous revision of the manuscript we demonstrated that the use of inhibitors such as EIPA, NaN₃ and Cytochalasin D causes disruption of the cell membrane that would compromise any method to measure endosomal escape (SLEEQ, FCS or otherwise).

2) The authors go to great length to discredit FCS as a valid technique, but do not include any experiments or literature citations to support their assertion “it is challenging to demonstrate that the area where the FCS measurement is taken is truly cytosolic”. FCS is low throughput, but it is the gold standard, and should be used by the authors to validate their assay (see below). Doing so would greatly improve this manuscript.

It was not our intention to discredit the FCS technique, and do not believe we have done so. However, we do believe it is important to highlight where the SLEEQ technique has advantages over FCS.

Our point that FCS measurements are subjective is based on the need to acquire signal from a single point within the cell. The selection of this point must be subjective as you need to exclude acquisition from the nucleus or from a point within the cell where there is endosomally trapped cargo. In the attached image from a paper where FCS was used, it is clear there is significant punctate endosomal signal that must be avoided. If care is not taken, the FCS signal measured could come from an endosome.

Taken from Figure 3a (ACS Cent Sci. 2018 4(10): 1379–1393.). Note the punctate fluorescence in the cytosol indicating the majority of material is in endosomes/lysosomes. The authors claim to have used FCS to detect 20nM concentration of material in the cytosol for this sample

The SLEEQ assay measures the signal from all cells in the well and there is no bias on which points are measured. These points are not meant to discredit FCS as a technique, just to highlight the limitations. We have revised the manuscript as follows to make this point clearer.

“Techniques such as FCS are also limited by a need to manually select a point in the cell to measure in the cytosolic concentration of fluorescently labelled protein. As the cytosol is a highly dynamic environment, it can be challenging to ensure that this point solely measures cytosolic material and excludes material from endosomal compartments that may traffic through the measurement point during the analysis period.”

We have also searched the literature for papers that use FCS to measure endosomal escape in live cells, and were only able to find 4 publications, all of which come from the same group. We don't think it is reasonable to assert that FCS is the gold standard based on this.

Another significant point that the reviewer has missed with SLEEQ vs FCS is that SLEEQ enables the escape efficiency to be measured. As is apparent from the image above (from the FCS paper), the majority of material remains trapped within the endosome, and FCS cannot measure the total concentration of cargo in the cells. Therefore it provides an incomplete understanding of what occurs with delivery to the cytosol.

3) The evidence provided to suggest that the delivered material is not degraded is insufficient. The conclusion drawn from the absence of HiBiT peptide is even worse (as the absence of a signal proves little – a classic “negative result”).

The reviewer appears to have misunderstood the additional data we included in Figure S8a. We have both a positive control for degradation (HiBiT only, far right lane) and a negative control (intact TAT-GFP-HiBiT, second from the right). The reviewer is incorrect to say we are drawing an inference from an absence of signal. In both of our sample lanes (0.5 hr and 4 hr) we see strong signal, but the signal we measure corresponds to intact TAT-GFP-HiBiT not to free HiBiT. Therefore we can conclude that signal we detect is from intact protein not degraded material.

In addition to Figure S8a, we have performed additional triplicate experiments to measure degradation of TAT-GFP-HiBiT over the 4 hour period of the experiment. Figure S8b shows

that the total signal from the cell lysate remains constant over the period of the experiment, demonstrating that there is no significant degradation of the protein.

4) No secondary validation

Again, the reviewer has missed the point of the paper, which is that there is no other assay capable of detecting endosomal escape efficiency, nor that is capable of detecting endosomal escape at the levels we can with SLEEQ.

The 'gold standard' for investigating endosomal escape is fluorescence microscopy, which we have performed (Fig 5 and S9). From the microscopy images, it is challenging to differentiate cytosolic signal from the punctate signal in the endosomes. This is a limitation with all microscopy based methods to determine endosomal escape, hence the need for the SLEEQ assay.

The results obtained by all other techniques we are aware of (including FCS and NanoClick) are limited to the analysis we perform in Figure 4a. They determine the amount of material delivered to the cytosol, and it is possible to compare the total amount of material delivered by one EEP compared to the other. None of these techniques look at the efficiency of delivery, and efficiency of delivery is essential to understanding what the EEPs are doing.

Furthermore, the sensitive of other techniques (including FCS and NanoClick) are not sufficient to detect endosomal escape at the physiologically relevant concentrations we have used. The concentration of sample used for measuring endosomal escape with FCS ranges from 1-10 μ M and can only detect ~20nM concentration of material in the cell. SLEEQ allows us to use physiologically relevant concentrations of material (down to 2nM) and to detect concentrations below 10 pM. This is more than 3 orders of magnitude lower than existing techniques.

5) The assay demands fusion of the protein being delivered to GFP. It is well known that cargo identity (in this case GFP) affects delivery.

This comment is incorrect and also misses the point of the paper. The assay does not demand fusion to GFP. HiBIT could be fused to any therapeutic protein of interest.

We agree that cargo identity affects delivery. The power of the SLEEQ assay and the results we present is that we measure delivery of the cargo and not just the ability of a small peptide to escape the endosome. For an EEP to be therapeutically relevant, it needs to enable the delivery of a cargo. Most papers (such as NanoClick mentioned in point 6 below) provide interesting information about the behaviour of the EEP, but not of whether the EEP can be used to deliver something other than the EEP itself.

There is a common misconception in the field that simply attaching an EEP to a protein can improve the efficiency of delivery to the cytosol. Our work here shows that while some EEPs do increase cytosolic delivery, they do this by increased total cell association and not improving the efficiency of escape.

6) Prior art (<https://pubs.acs.org/doi/10.1021/acscchembio.0c00804>). This assay uses an azide tag (N3 – far less perturbing than GFP) and is just as sensitive.

The reviewer is incorrect to say the technique is as sensitive as SLEEQ. The sensitivity of the NanoClick assay is ~50nM (similar to FCS), the sensitivity of SLEEQ is <10 pM (3 orders of magnitude lower).

The NanoClick paper showcases an interesting technique, however as highlighted above, like many other papers does not measure the efficiency of endosomal escape. It does not test if the EEPs can promote delivery of anything other than the EEP itself, and it does not exclude the possibility that the signal which is measured is coming from dye that has been cleaved from the peptide.

We also note this paper has been submitted and published after the submission of this manuscript. However, we do feel that it highlights the interest in understanding endosomal escape and the importance of the work in our paper. We have included the paper as reference 38 in the revised manuscript.

Reviewers' Comments:

Reviewer #1:

Remarks to the Author:

The authors have addressed my comments. I believe that the revised manuscript has improved by providing experimental controls as well as nuance about what is a complicated/confusing/debated process overall (how CPPs interact with cells). Jean-Philippe Pellois